# On the Recoverability of Causal Relations from Temporally Aggregated I.I.D. Data

## Abstract

Real-world data in fields such as economics, finance and neuroscience often exhibit a lower resolution compared to the underlying causal process, with temporally aggregated data being a common example. While the impact of temporally aggregated time series on temporal causal discovery has received attention, the effects of highly aggregated data, which yield independent and identically distributed (i.i.d.) observations, on instantaneous (non-temporal) causal discovery have been largely overlooked by the research community. There is substantial evidence suggesting that temporally aggregated i.i.d. data are prevalent in reality. This prevalence arises because the time required for causal interactions is often considerably shorter than the observational interval, leading to a large aggregation factor and subsequently rendering the temporally aggregated data i.i.d. The critical question arises: are causal discovery results obtained from such data consistent with the true causal process? In this paper, we provide theoretical conditions necessary to ensure the consistency of causal discovery results when analyzing temporally aggregated i.i.d. data. Through a combination of theoretical analysis and experimental validation, we demonstrate that conducting causal discovery on such data often leads to erroneous results. Our primary objective is to bring attention to the risks associated with performing causal discovery on highly aggregated i.i.d. data and advocate for a cautious and meticulous approach when interpreting causal discovery outcomes derived from such data.

## 1 Introduction

Causal discovery methods, which aim to uncover causal relationships from observational data, have been extensively researched and utilized across multiple disciplines including computer science, economics, and social sciences (Pearl, 2009; Spirtes et al., 2000). These methods can be broadly categorized into two types. The first type is temporal causal discovery, which is specifically designed for analyzing time series data. Examples of this type include the Granger causality test (Granger, 1969) and its variants. The second type is instantaneous (non-temporal) causal discovery, which is applicable to independent and identically distributed (i.i.d.) data. This category encompasses various approaches such as constraint-based, score-based, and functional causal model (FCM)-based methods like PC (Spirtes et al., 2000), GES (Chickering, 2002), and LiNGAM (Shimizu et al., 2006). All these methods, whether temporal or non-temporal, are premised on the assumption that the causal frequency aligns with the observation frequency.

In real-world scenarios, the causal frequency is often unknown, which means that the available observations may have a lower resolution than the underlying causal process. An instance of this is annual income, which is an aggregate of monthly or quarterly incomes (Drost & Nijman, 1993). Furthermore, it is widely believed that causal interactions occur at high frequencies in fields such as economics (Ghysels et al., 2016) and neuroscience (Zhou et al., 2014). Extensive research has been conducted to explore the effects of temporal aggregation on time series modeling (Ghysels et al., 2016; Marcellino, 1999; Silvestrini & Veredas, 2008; Granger & Lee, 1999; Rajaguru & Abeysinghe, 2008). These works typically consider small aggregation factor k[1] and still treat the temporal aggregation from causal processes as a time series.

---

[1]The "aggregation factor $k$" refers to the number of data points from the underlying causal process that are combined to form each observed data point. It is also called the aggregation level or aggregation period.

However, in many real-world scenarios, the temporal aggregation factor can be quite large. In these cases, what was originally a time-delayed causal relationship can appear as an instantaneous causal relationship when observed. Data that was originally time-series can also become i.i.d. data. An example commonly used in statistics and econometrics textbooks to illustrate correlation, causation, and regression analysis is the influence of temperature on ice cream sales. Intuitively, one might think that the average daily temperature has an instantaneous causal effect on the total daily ice cream sales. However, in reality, the causal process involves a time lag: a high temperature at a specific past moment influences people's decision to purchase ice cream, which then leads to a sales transaction at a subsequent moment. Unfortunately, we often lack access to these precise moment-to-moment details. Instead, we typically work with temporally aggregated data, such as the average daily temperature and the total daily ice cream sales, which represent the sum of all individual sales transactions over the day. As a result of this aggregation, the original time-delayed causal relationship becomes an instantaneous causal relationship when observed.

Interestingly, the causality community has long acknowledged the significance of temporal aggregation as a common real-world explanation for instantaneous causal models like the structural equation model. Fisher (1970) argued that simultaneous equation models serve as approximations of true time-delayed causal relationships driven by temporal aggregation in the limit. He emphasized that while causation inherently involves a temporal aspect, as the reaction interval tends to zero, the aggregation factor k tends to infinity. Granger (1988) shared a similar view and claimed that "temporal aggregation is a realistic, plausible, and well-known reason for observing apparent instantaneous causation". This explanation has been consistently used in recent causal discovery papers, especially those discussing cyclic models (Rubenstein et al., 2017; Lacerda et al., 2012; Hyttinen et al., 2012).

When applying causal discovery methods to uncover instantaneous causal relationships resulting from temporal aggregation, a fundamental question arises: Are these instantaneous causal relationships consistent with the true time-delayed causal relationships? This issue is crucial because our primary concern lies in discerning the true causal relations. If the results obtained by the instantaneous causal discovery methods do not align with the true causal relationship, the results will hold no meaningful value. Regrettably, few studies have examined the alignment of these "spurious" instantaneous causal relationships stemming from temporal aggregation with the true time-delayed causal relationships. The only theoretical analysis we could find related to this question is given by Fisher (1970) and Gong et al. (2017). We will delve into a comprehensive discussion of their contributions in section 2.

In this paper, we primarily investigate under what conditions and to what extent we can recover true causal information from temporally aggregated data. The primary aim is to alert the community to the potential impact of temporal aggregation on the results of non-temporal causal discovery when analyzing real-world data. Since we aim to recover the causal structure of the true causal process from temporally aggregated data, we hope that the aggregated data maintain some consistency with the true causal process. We categorize this consistency into functional consistency and conditional independence consistency, which respectively correspond to the recoverability of FCM-based and constraint-based causal discovery methods. For functional consistency, we find it difficult to hold in the nonlinear case. Even in the linear non-Gaussian case, functional-based causal discovery loses its identifiability because temporal aggregation will change non-Gaussian noise to Gaussian noise. As for conditional independence consistency, it cannot be guaranteed in the general case either. However, it is less strict than functional consistency because partial linearity is sufficient for it.

## 2 RELATED WORK

Fisher (1970) established the corresponding relationship between the simultaneous equation model and the true time-lagged model, providing the conditions to ensure such correspondence. His analysis encompassed both linear and general cases. Roughly speaking, he conducted theoretical analysis to show that this correspondence can be ensured when the function of the equation has a fixed point. However, the assumptions he employed were quite restrictive, assuming that the value of noise is fixed for all the causal reactions during the observed interval. Some subsequent studies have also adopted this assumption (Rubenstein et al., 2017; Lacerda et al., 2012). This assumption is too strong, as it implies that noise in the causal reaction is only related to our observation. Actually, the noise defined in structural causal models or functional causal models also represents unobserved or

unmeasured factors that contribute to the variation in a variable of interest, rather than being merely observational noise.

Gong et al. (2017) adopted a more reasonable and flexible assumption in their work. They defined the original causal process as a vector auto-regressive model (VAR) $X_t = AX_{t-1} + e_t$ and allowed for the randomness of the noise $e_t$ in the observation interval. They gave a theoretical analysis showing that, in the linear case and as the aggregation factor tends to infinity, the temporally aggregated data $\overline{X}$ becomes i.i.d. and compatible with a structural causal model $\overline{X} = A\overline{X} + \overline{e}$. In this model, matrix A is consistent with the matrix A in the original VAR. This suggests that high levels of temporal aggregation preserve functional consistency in the linear case. However, their study only considers the linear case and lacks analysis for general cases.

To the best of our knowledge, our paper is the first to specifically discuss the risks and feasibility of performing instantaneous causal discovery methods on temporally aggregated data in general cases.

## 3 FUNCTIONAL CONSISTENCY: RECOVERABILITY OF FCM-BASED METHODS

FCM-based methods make stronger assumptions and utilize more information beyond just conditional independence. Thus, they can distinguish cause from effect from observational data under the functional assumptions. If we want to ensure the reliability of the results from FCM-based causal discovery on temporally aggregated data, we need some functional consistency between the process of temporally aggregated data and the true causal process.

### 3.1 DEFINITIONS AND PROBLEM FORMULATION

In this section, aligning with the settings in Gong et al. (2017), our research focuses on the general VAR(1) model $X_t = f(X_{t-1}, e_t)$, which serves as the underlying causal process. More specifically, we assume the underlying causal process can be described by a VAR(1):

$$X_t = f(X_{t-1}, e_t), \quad t \geq 2,$$

where $X_t = (X_t^{(1)}, X_t^{(2)}, \ldots, X_t^{(s)})^T$ is the observed data vector at time $t$, $s$ is the dimension of the random vector. $f$ is a vector-valued function $\mathbb{R}^{2s} \to \mathbb{R}^s$, and $e_t = (e_t^{(1)}, \ldots, e_t^{(s)})^T$ denotes a temporally and contemporaneously independent noise process. When mentioning VAR in our paper, we refer to the general VAR model defined above, which includes both linear and nonlinear functions. The initial data vector $X_1$ is assumed to follow a distribution with independent components.

The temporally aggregated time series of this process, denoted by $\overline{X_t}$, is defined as:

$$\overline{X_t} = \frac{\sum_{i=1}^k X_{i+(t-1)k}}{g(k)}. \tag{1}$$

In this paper, we only consider cases where k is large. In such cases, we treat the temporally aggregated data as i.i.d. data and we will drop the subscript $t$ in Eq. 1 from now on. g(k) generally requires $\lim_{k \to \infty} g(k) = +\infty$, like g(k)=k, but when discussing aggregation of instantaneous causal model and k is finite, the choice of g(k) doesn't matter, it can also be g(k)=1.

**Definition 1** (Functional Consistency). *Consider an underlying causal process generating temporally aggregated data. This process is said to exhibit functional consistency if there exists a function $\hat{f}$ such that for any realization of the states $X_{1:k}$, and the independent noises encountered in the process $e_{2:k+1}$, the temporally aggregated data $\overline{X}$ satisfies the equation $\overline{X} = \hat{f}(\overline{X}, e)$. Here, $e$ denotes a noise vector comprising independent components only depend on $e_{2:k+1}$.*

This definition implies that if functional consistency holds, then the aggregated data can at least be represented as a Structural Causal Model (SCM) in vector form, and the source of independent noise aligns with the underlying process. Please note that we allow the generative mechanism $\hat{f}$ to differ from the underlying causal function $f$. However, even with this allowance, achieving functional consistency in the nonlinear case remains challenging, as we will demonstrate in this section.

Following this definition, we will provide answers to this question: Under what conditions can we ensure functional consistency for temporal aggregation? According to the definition of temporal aggregation:

$$\overline{X} = \frac{1}{g(k)}\sum_{i=2}^{k+1} X_i + \frac{X_1 - X_{k+1}}{g(k)} = \frac{1}{g(k)}\sum_{i=1}^{k} f(X_i, e_{i+1}) + \frac{X_1 - X_{k+1}}{g(k)}.$$

When $k$ becomes larger(see Appendix F for what k value is large enough in practice), the second term $\frac{X_1 - X_{k+1}}{g(k)}$ will tend to 0. Thus, we will mainly consider under which conditions we can have:

$$\frac{1}{g(k)}\sum_{i=1}^{k} f(X_i, e_{i+1}) = \hat{f}(\frac{1}{g(k)}\sum_{i=1}^{k} X_i, e) = \hat{f}(\overline{X}, e)$$

holds for some $e$ with independent components. The linear case is straightforward, which is already solved by Fisher (1970); Gong et al. (2017). This is because in linear case $\frac{1}{g(k)}\sum_{i=1}^{k} f(X_i, e_{i+1}) = \frac{\sum_{i=1}^{k}(AX_i + e_{i+1})}{\sqrt{k}} = A\overline{X} + \overline{e}$.

## 3.2 FINITE K

The choice of $g(k)$ is linked to the random process's asymptotic behavior. Since we do not make any assumptions about distribution on $X_t$ here, we cannot determine $g(k)$ or establish the limit of temporal aggregation, because several elements are not well-defined, making it difficult to ascertain its convergence. Therefore, we have to conduct the analysis in the finite case first. If this equation holds in the finite case, it will definitely hold in the infinite case.

We then arrive at the following theorem:

**Theorem 1.** *Consider a function $f(X, e)$ that is differentiable with respect to $X$. Define the following:*

*Statement 1: The function $f$ is of the form $f(x, e) = Ax + f_2(e)$ for some function $f_2$.*

*Statement 2: For any positive integer $k$, there exists a function $\hat{f}$ such that the functional equation $\frac{\sum_{i=1}^{k} f(X_i, e_{i+1})}{g(k)} = \hat{f}(\overline{X}, e)$ holds for any $X_i$, $e_i$, and any normalization factor $g(k)$, where $e$ is related only to $e_i$ for $i = 2, \ldots, k + 1$.*

*Statement 1 is a necessary condition for Statement 2.*

See Appendix A for the proof. From this theorem, we realize that it is very difficult to relax the linearity assumption to ensure functional consistency. It means that if the underlying process is nonlinear, applying nonlinear FCM-based causal discovery methods on the aggregated data cannot ensure correct discovery.

## 3.3 INFINITE K

We have demonstrated that when the model is linear, the time-delay function can be preserved in simultaneous equations. And from the theorem 1, we know relaxing this linearity to ensure consistency with finite k is highly challenging. Yet, we are going to present a more negative result: even in the linear non-Gaussian case, which perfectly fits the requirements of many function-based causal discovery methods (Shimizu et al., 2006), we still cannot guarantee identifiability when $k$ is large.

Consider the linear model that preserves the functional structure:$\overline{X} = A\overline{X} + \overline{e}$. When $e_i$ is non-Gaussian, we can identify the adjacency matrix $A$ for finite values of $k$. This is due to the fact that $\overline{e} = \frac{\sum_{i=1}^{k} e_i}{\sqrt{k}}$ remains non-Gaussian. However, as $k$ increases, $\overline{e}$ will converge to a Gaussian distribution as a consequence of the central limit theorem. Consequently, the model becomes linear Gaussian, rendering it unidentifiable. Our simulation experiments in 5.1 demonstrate that the model becomes unidentifiable rapidly as $k$ increases.

# 4 CONDITIONAL INDEPENDENCE CONSISTENCY: RECOVERABILITY OF CONSTRAINT-BASED METHOD

Constraint-based causal discovery methods utilize conditional independence to identify the Markov equivalence classes of causal structures. These methods heavily rely on the faithfulness assumption, which posits that every conditional independence in the data corresponds to a d-separation in the underlying causal graph.

The information utilized by constraint-based causal discovery methods is less than that used by FCM-based methods. This implies that the consistency we require for the recoverability of constraint-based methods on temporally aggregated data is less stringent than functional consistency. In essence, we only need the temporally aggregated data to preserve the conditional independence of the summary graph of the underlying causal process. If the temporal aggregation maintains such conditional independence consistency, then the constraint-based causal discovery method can recover the Markov equivalence class of the summary graph entailed by the underlying true causal process.

## 4.1 DEFINITIONS AND PROBLEM FORMULATION

To examine whether the temporally aggregated data preserves the conditional independence consistency with the summary graph of the underlying causal process, we will discuss the three fundamental causal structures of the summary graph: the chain, the fork, and the collider. We will provide theoretical analysis for each of these three fundamental cases respectively.

In subsection 3.1, we assume the original causal process is VAR(1) and we work with the temporal aggregation of it. But in this section, for analytical convenience we will assume the original model is an aligned version of VAR(1), and work with the temporal aggregation of it. We will show this alignment is reasonable because the temporal aggregation of these two original processes is the same when $k$ is large.

### 4.1.1 ALIGNED MODEL

In the true causal process, all the causal effects between different components are cross-lagged effects from the previous state of components to the current state of other components.

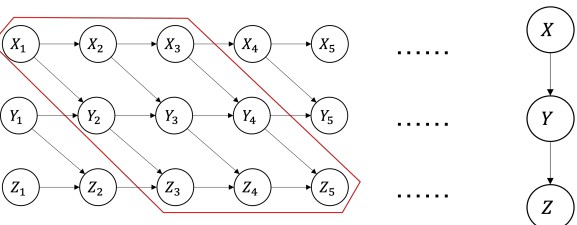

Figure 1: Left: Directed acyclic graph for the VAR model with chain-like cross lag effects. Right: The corresponding summary graph.

See Figure 1 as an example. The summary graph of this VAR is a chain structure involving triple variables. But the chain structure actually occurs in a lagged form: $X_t \to Y_{t+1} \to Z_{t+2}$[2]. For analytical convenience, we will perform an alignment $X'_t := X_t, Y'_t = Y_{t+1}, Z'_t := Z_{t+2}$ to make the causal effect instantaneous. We refer to this as the aligned model. In the example of chain-like VAR, the aligned model is in the red box in the Figure 1.

While instantaneous causal relationships are considered unlikely in the real world, this alignment is reasonable for theoretical analysis. This is because our focus is actually on the temporally aggregated data. When $k$ is large, the temporal aggregation from the original VAR and the aligned

---

[2]Here $X_t$ represents a one-dimensional variable. Starting from this section, $X_t$ and $\overline{X}$ represent a one-dimensional variable, and we will use X, Y, Z,... to represent the different components of multivariate time series, instead of using $X^{(1)},...,X^{(s)}$ as defined in 3.1.

model is exactly the same: as $k$ approaches infinity, the temporally aggregated data $(\overline{X'}, \overline{Y'}, \overline{Z'})$ tends towards $(\overline{X}, \overline{Y}, \overline{Z})$. This can be demonstrated by the following equalities: Since $g(k) \to \infty$, we have

$$\overline{Y'} - \overline{Y} = \frac{\sum_{i=2}^{k+1} Y_i}{g(k)} - \frac{\sum_{i=1}^{k} Y_i}{g(k)} = \frac{Y_{k+1} - Y_1}{g(k)} \to 0,$$

$$\overline{Z'} - \overline{Z} = \frac{\sum_{i=3}^{k+2} Z_i}{g(k)} - \frac{\sum_{i=1}^{k} Z_i}{g(k)} = \frac{Z_{k+2} + Z_{k+1} - Z_1 - Z_2}{g(k)} \to 0,$$

as $k \to \infty$.

**Definition 2** (Aligned Model with Instant Structures). *The aligned model for VAR model with structure function $f_X, f_Y, f_Z$ incorporating chain-like cross lag effect is given by:*

*$X_0$, $Y_0$, $Z_0$ are independent and follow the initial distribution. when $t \geq 1$,*

**Chain-like Model:** $X_t = f_X(X_{t-1}, e_{X,t})$, $Y_t = f_Y(X_t, Y_{t-1}, e_{Y,t})$, $Z_t = f_Z(Y_t, Z_{t-1}, e_{Z,t})$,

**Fork-like Model:** $X_t = f_X(X_{t-1}, Y_t, e_{X,t})$, $Y_t = f_Y(Y_{t-1}, e_{Y,t})$, $Z_t = f_Z(Y_t, Z_{t-1}, e_{Z,t})$,

**Collider-like Model:** $X_t = f_X(X_{t-1}, e_{X,t})$, $Y_t = f_Y(X_t, Y_{t-1}, Z_t, e_{Y,t})$, $Z_t = f_Z(Z_{t-1}, e_{Z,t})$,

*where $f_X, f_Y, f_Z$ are general functions. $e_{X,t}, e_{Y,t}, e_{Z,t}$ are independent random variables with non-zero variance, which are independent of each other and they are identically distributed and independent across time t.*

*The temporal summation and aggregation are denoted as $S_X := \sum_{i=1}^{k} X_i$, $\overline{X} := \frac{S_X}{g(k)}$, and similarly for $S_Y$, $\overline{Y}$, $S_Z$, and $\overline{Z}$. When $k$ is finite, $S_X$, $S_Y$, and $S_Z$ have the same conditionally independent relationship with $\overline{X}$, $\overline{Y}$, and $\overline{Z}$, respectively. Therefore, for simplicity, we analyze $S_X$, $S_Y$, and $S_Z$ when $k$ is finite.*

The figures of the aligned models of three fundamental structure involving temporal aggregation variables are presented in Figure 2.

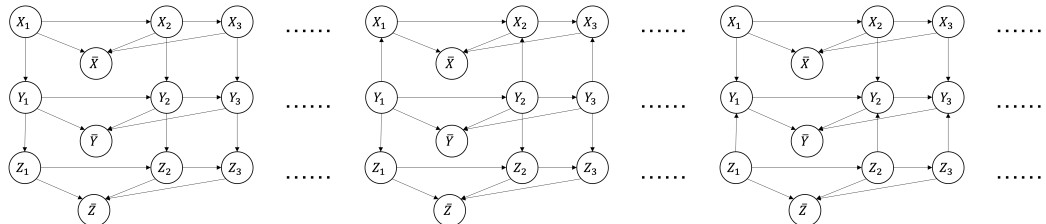

Figure 2: Left: Chain-like aligned model. Center: Fork-like aligned model. Right: Collider-like aligned model.

### 4.1.2 PROBLEM FORMULATION

**Definition 3** (Summary Graph). *Aligned with Peters et al. (2013); Gong et al. (2023), each time series component is collapsed into a node to form the summary causal graph. The summary graph represents causal relations between time series without referring to time lags. If there is an arrow from $X_{i,t-k}$ to $X_{j,t}$ in the original process for some $k \geq 0$, then it is represented in the summary graph.*

**Definition 4** (Conditional Independence Consistency). *Consider an underlying causal process generating temporally aggregated data. This process is said to exhibit conditional independence consistency if the distribution of temporally aggregated data entails a conditional independence set that is consistent with the d-separation set in the summary graph entailed by the original process.*

We will address the problem of determining the conditions under which temporal aggregation preserves conditional independence consistency in three fundamental causal structures: chain, fork, and collider.

## 4.2 NECESSARY AND SUFFICIENT CONDITIONS FOR CONSISTENCY

For the figure of a chain or fork structure, we expect the middle node can d-separate the nodes on both sides. However, from the structure of Figure 2 Left and Center, we can find that all adjacent nodes of $\overline{Y}$ point to $\overline{Y}$. Therefore, when we condition on $\overline{Y}$, we cannot block any path.

For the figure of a collider structure, we expect the nodes on both sides are unconditionally independent and when conditioned on the middle node, the nodes on both sides will be dependent. Fortunately, from the structure of Figure 2 Right, we can find that all the $Y_t$ are collider for $X_t$ and $Z_t$ so $\overline{X} \perp\!\!\!\perp \overline{Z}$ unconditionally. And because $\overline{Y}$ is a descendant of these colliders, when we condition on $\overline{Y}$, the path involving $X_t$ and $Z_t$ will be open. As a result, $\overline{X}$ is dependent with $\overline{Z}$ conditional on $\overline{Y}$.

**Remark 1** (Conditional Independence Consistency under Faithfulness Condition). *Assume the aligned models satisfy the causal Markov condition and causal faithfulness condition.*

- *The conditional independent sets of temporal aggregation of chain-like/fork-like aligned model is $\emptyset$, which is **not** consistent with the chain/fork structure.*

- *The conditional independent sets of temporal aggregation of collider-like aligned model is $\overline{X} \perp\!\!\!\perp \overline{Z}$, which is consistent with the collider structure.*

This remark emphasizes that in general cases, the temporal aggregation of a model with chain-/fork-like structure does not exhibit the same conditional independence as a genuine chain or fork structure under the faithfulness assumption. As a result, we will explore the conditions required to ensure the validity of the conditional independence $\overline{X} \perp\!\!\!\perp \overline{Z} \mid \overline{Y}$ in the context of temporal aggregation.

**Theorem 2** (Necessary and Sufficient Condition for Conditional Independence Consistency of Chain and Fork Structure). *Consider the distribution of $(S_X, S_Y, S_Z, Y_1, ..., Y_k)$ entailed from the aligned model, the following statements are equivalent when $2 \leq k < \infty$:*

*(i) Conditional Independence: $S_X \perp\!\!\!\perp S_Z \mid S_Y$*

*(ii) Conditional Probability Relation: $\forall s_X, s_Y, s_Z \in \mathbb{R}$*

$$\iint_{\mathbb{R}^k} \alpha(s_Z, s_Y, y_{1:k}) \left( \beta(y_{1:k}, s_Y, s_X) - \gamma(y_{1:k}, s_Y) \right) dy_1 ... dy_k = 0 \tag{2}$$

*(iii) Alternative Conditional Probability Relation: $\forall s_X, s_Y, s_Z \in \mathbb{R}$*

$$\iint_{\mathbb{R}^k} \alpha^*(s_X, s_Y, y_{1:k}) \left( \beta^*(y_{1:k}, s_Y, s_Z) - \gamma(y_{1:k}, s_Y) \right) dy_1 ... dy_k = 0 \tag{3}$$

*where*

- $\alpha(s_Z, s_Y, y_{1:k}) := p_{S_Z|S_Y,Y_{1:k}}(s_Z|s_Y, y_{1:k})$

- $\beta(y_{1:k}, s_Y, s_X) := p_{Y_{1:k}|S_Y,S_X}(y_{1:k}|s_Y, s_X)$

- $\alpha^*(s_X, s_Y, y_{1:k}) := p_{S_X|S_Y,Y_{1:k}}(s_X|s_Y, y_{1:k})$

- $\beta^*(y_{1:k}, s_Y, s_X) := p_{Y_{1:k}|S_Y,S_X}(y_{1:k}|s_Y, s_X)$

- $\gamma(y_{1:k}, s_Y) := p_{Y_{1:k}|S_Y}(y_{1:k}|s_Y)$

See Appendix B for the proof. From this sufficient and necessary condition, we find that the integrand can be divided into two parts. For example, the integrand in formula 2 can be divided into two parts. The first part is $p_{S_Z|S_Y,Y_{1:k}}(s_Z|s_Y, y_{1:k})$. Because $Y_1, ... Y_k$ d-separate $S_Z$ from $X_1, ..., X_k$ perfectly, this part is related to the causal mechanism between Y and Z. The second part is $\left( p_{Y_{1:k}|S_Y,S_X}(y_{1:k}|s_Y, s_X) - p_{Y_{1:k}|S_Y}(y_{1:k}|s_Y) \right)$ is related to the causal mechanism between Y and Z. This inspires us to consider different parts of the model individually.

**Corollary 1** (Sufficient Conditions for Conditional Independence). *If $\{S_X \perp\!\!\!\perp Y_{1:k} \mid S_Y\}$ **or** $\{S_Z \perp\!\!\!\perp Y_{1:k} \mid S_Y\}$ holds, then $S_X \perp\!\!\!\perp S_Z \mid S_Y$ holds.*

Proof can be found in Appendix B. This corollary has a very intuitive interpretation: when the information needed to infer $S_X$ from $Y_{1:k}$ is completed included in $S_Y$, then conditioning on $S_Y$ is

equivalent to conditioning on $Y_{1:k}$. In this case, $Y_{1:k}$ d-separate $S_X$ from $S_Z$. The same principle applies to $S_Z$. When does the information to infer $S_X/S_Z$ from $Y_{1:k}$ is completely included in $S_Y$?

**Corollary 2** (Partial Linear Conditions)**.**

1. *For a fork-like aligned model:*

   - *If $f_Z(Y_t, Z_{t-1}, e_{Z,t})$ is of the form $\alpha * Y_t + e_t$, where $\alpha$ can be any real number, then $S_Z \perp\!\!\!\perp Y_{1:k} \mid S_Y$.*
   - *If $f_X(X_{t-1}, Y_t, e_{X,t})$ is of the form $\alpha * Y_t + e_t$, where $\alpha$ can be any real number, then $S_X \perp\!\!\!\perp Y_{1:k} \mid S_Y$.*

2. *For a chain-like aligned model:*

   - *If $f_Z(Y_t, Z_{t-1}, e_{Z,t})$ is of the form $\alpha * Y_t + e_t$, where $\alpha$ can be any real number, then $S_Z \perp\!\!\!\perp Y_{1:k} \mid S_Y$.*
   - *If the time series is stationary and Gaussian, and $f_Y(X_t, Y_{t-1}, e_{Y,t})$ is of the form $\alpha * X_t + e_t$, where $\alpha$ can be any real number, then $S_X \perp\!\!\!\perp Y_{1:k} \mid S_Y$.*

Proof can be found in Appendix B. Roughly speaking, this corollary suggests that if the causal relationship between X/Z and Y is linear, then the information needed to infer $S_X/S_Z$ from $Y_{1:k}$ is completely included in $S_Y$. Further, based on the sufficient condition for conditional independence (refer to Corollary 1), we can see that it is not necessary for the entire system to be linear.

## 5   Simulation Experiments

We conducted five experiments to comprehensively address the various aspects of the aggregation problem. Firstly, we applied widely-used causal discovery methods(PC(Peters et al., 2013), FCI(Peters et al., 2013), GES(Chickering, 2002)) to aggregation data with 4 variables, enabling readers to grasp the motivation and core issue discussed in this paper. Secondly and thirdly, we conducted experiments on functional consistency(apply Direct LiNGAM(Shimizu et al., 2011)/ANM(Hoyer et al., 2008) to linear/nonlinear data with different aggregation levels) and conditional independence consistency(perform Kernel Conditional Independence test(Zhang et al., 2012) on aggregated data) to bolster the theorems presented in the main text. Fourthly, we carried out an experiment to investigate the impact of the k value and to justify the approximations made in this paper. Fifthly, we performed the PC algorithm with a skeleton prior on aggregated data and consistently obtained correct results, offering a preliminary solution to the aggregation problem and laying the groundwork for future research in this area. Due to the page limit, we present only a limited number of results in the main text. Detailed settings and results of the five experiments can be found respectively in the Appendices: C, D, E, F, and G.

### 5.1   FCM-based Causal discovery in Linear non-Gaussian case

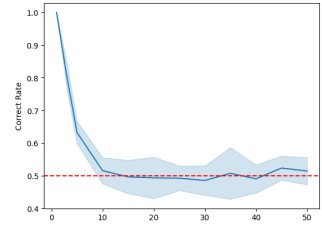

Here we examine the use of a FCM-based causal discovery method on bivariate temporally aggregated data in the linear non-Gaussian case to distinguish between cause and effect. Specifically, we employ the Direct LiNGAM method to represent FCM-based causal discovery methods.

We perform a simulation experiment on the model $Y_i = 2X_i + e_{Y,i}$, where the noise follow uniform distribution. We then generating the dataset $\overline{X}, \overline{Y}$ with a sample size of 10,000. We apply the Direct LiNGAM method on this dataset to determine the causal order.

Figure 3: Relationship between the aggregation factor $k$ and the performance of the Direct LiNGAM method.

To investigate the relationship between the aggregation factor $k$ and the performance of the Direct LiNGAM method, we vary the values of $k$ from 1 to 100 to see the correct rate in 100 repetitions.

Our results indicate that when $k$ is small, the correct rate is near 100%, implying a good performance of the Direct LiNGAM method. However, as $k$ increases from 3 to 30, the correct rate drops rapidly to 50% as random guess. This experiment demonstrates that

even in the linear non-Gaussian case, temporal aggregation can significantly impair the identifiability of functional-based methods relying on non-Gaussianity.

## 5.2 CONDITIONAL INDEPENDENCE TEST IN GAUSSIAN CASE

We perform experiments using three different structures: chain, fork, and collider to validate our theoretical results.

In all structures, each noise term $e$ follows an independent and identically distributed (i.i.d) standard Gaussian distribution. For the causal relationship between $X$ and $Y$, we use the function $f(\cdot, e)$. In the linear case, $f(\cdot) = (\cdot)$. In the nonlinear case, we use the post-nonlinear model $f(\cdot, e) = G(F(\cdot) + e)$ (Zhang & Hyvarinen, 2012) and uniformly randomly pick $F$ and $G$ from $(\cdot)^2$, $(\cdot)^3$, and $tanh(\cdot)$ for each repetition. This is to ensure that our experiment covers a wide range of nonlinear cases. Similarly, the same approach is applied for the relationship between $Y$ and $Z$ with the corresponding function $g(\cdot)$.

We set t=1,2, $S_X = X_1 + X_2$, $S_Y = Y_1 + Y_2$, $S_Z = Z_1 + Z_2$. And we generate 1000 i.i.d. data points for $(X_1, X_2, Y_1, Y_2, S_X, S_Y)$ and feed them into the approximate kernel-based conditional independence test (Strobl et al., 2019). We test the null hypothesis(conditional independence) for (I) $S_X \perp\!\!\!\perp S_Y$, (II) $S_Y \perp\!\!\!\perp S_Z$, (III) $S_X \perp\!\!\!\perp S_Z$, (IV) $S_X \perp\!\!\!\perp S_Y \mid S_Z$, (V) $S_Y \perp\!\!\!\perp S_Z \mid S_X$, (VI) $S_X \perp\!\!\!\perp S_Z \mid S_Y$. We also test for the conditional independence in corollary 1: (A) $S_X \perp\!\!\!\perp Y_1 \mid S_Y$, and (B) $S_Z \perp\!\!\!\perp Y_1 \mid S_Y$. We report the rejection rate for fork structure at a 5% significance level in 100 repeated experiments in Table 2b. The results for chain structure and collider structure can be found in Appendix 2a.

Table 1: Rejection rates for CIT tests with different combinations of linear and nonlinear relationships. The index in the box represents the conditional independence that the structure should ideally have. Simply speaking, for the column VI, the closer the rejection rate is to 5%, the better. For all other columns from I to V, a higher rate is better.

| $X_t \to Y_t$ | $Y_t \to Z_t$ | I | II | III | IV | V | VI | A | B |
|---|---|---|---|---|---|---|---|---|---|
| Linear | Linear | 100% | 100% | 100% | 100% | 100% | 5% | 6% | 5% |
| Nonlinear | Linear | 92% | 100% | 84% | 92% | 100% | 5% | 76% | 5% |
| Linear | Nonlinear | 100% | 93% | 85% | 100% | 93% | 5% | 5% | 71% |
| Nonlinear | Nonlinear | 92% | 93% | 72% | 86% | 87% | 58% | 72% | 74% |

The experiment shows that as long as there is some linearity, we can find a consistent conditional independence set, which aligns with Corollary 2. However, in completely non-linear situations, we still cannot find a consistent conditional independence set, which aligns with Remark 1.

## 6 CONCLUSION AND LIMITATION

This paper points out that although many people use the occurrence of instantaneous causal relationships due to temporal aggregation as a real-world explanation for instantaneous models, few people pay attention to whether these instantaneous causal relationships are consistent with the underlying time-delayed causal relationships when this situation occurs. This paper mainly discusses whether the causal models generated by temporal aggregation maintain functional consistency and conditional independence consistency in general (nonlinear) situations. Through theoretical analysis in the case of finite $k$, we show that functional consistency is difficult to achieve in non-linear situations. Furthermore, through theoretical analysis and experimental verification in the case of infinite k, we show that even in linear non-Gaussian situations, the instantaneous model generated by temporal aggregation is still unidentifiable. For conditional independence consistency, we show through sufficient and necessary conditions and experiments that it can be satisfied as long as the causal process has some linearity. However, it is still difficult to achieve in completely non-linear situations.

Limitations: Although the negative impact of temporal aggregation on instantaneous causal discovery has been pointed out, a solution has not been provided.

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

## A   PROOF FOR NECESSARY CONDITION OF FUNCTIONAL CONSISTENCY

**Theorem.** *Consider a function $f(X, e)$ that is differentiable with respect to $X$. Define the following:*

*Statement 1: The function $f$ is of the form $f(x, e) = Ax + f_2(e)$ for some function $f_2$.*

*Statement 2: For any positive integer $k$, there exists a function $\hat{f}$ such that the functional equation $\frac{\sum_{i=1}^{k} f(X_i, e_{i+1})}{g(k)} = \hat{f}(\overline{X}, e)$ holds for any $X_i$, $e_i$, and any normalization factor $g(k)$, where $e$ is related only to $e_i$ for $i = 2, \ldots, k+1$.*

*Statement 1 is a necessary condition for Statement 2.*

*Proof.* Suppose the functional equation

$$\frac{\sum_{i=1}^{k} f(X_i, e_{i+1})}{g(k)} = \hat{f}(\overline{X}, e) \tag{4}$$

holds, where $e$ is dependent only on $\{e_i\}_{i=2}^{n+1}$. Setting $k = 2$, we have $f(X, e_1) + f(\mathbf{0}, e_2) = \hat{f}(X, e)$ and $f(\mathbf{0}, e_1) + f(X, e_2) = \hat{f}(X, e)$ and please note e is the same in both equations because e is only related to $e_1, e_2$. Then we obtain:

$$f(X, e_1) + f(\mathbf{0}, e_2) = f(\mathbf{0}, e_1) + f(X, e_2) \tag{5}$$

for all $X$, $e_1$, and $e_2$.

Given that $f(X, e)$ is differentiable with respect to $X$, we can differentiate both sides of the equation to yield:

$$\frac{\partial f(X, e_1)}{\partial X} = \frac{\partial f(X, e_2)}{\partial X}. \tag{6}$$

Due to the arbitrariness of $e_1$ and $e_2$, this implies that $\frac{\partial f(X,e)}{\partial X}$ is independent of $e$. Hence, it follows that $f$ must have the form: $f(X, e) = f_1(X) + f_2(e)$ for some functions $f_1$ and $f_2$. Let's assume $f_1(\mathbf{0}) = 0$(this is always achievable because the constant term $f_1(\mathbf{0})$ can be incorporated into $f_2$).

Let $k = 2$, and $g(2) = 1$. Then, for any $X_1, e_1, X_2, e_2$, we have:

$$f(X_1, e_1) + f(X_2, e_2) = f_1(X_1) + f_1(X_2) + f_2(e_1) + f_2(e_2) \tag{7}$$

And due to equation 4, we have:

$$f(X_1, e_1) + f(X_2, e_2) = f_1(X_1) + f_1(X_2) + f_2(e_1) + f_2(e_2) = \hat{f}(X_1 + X_2, e) \tag{8}$$

Because $\hat{f}(X_1 + X_2, e) - f_1(e_1) - f_2(e_2) = f_1(X_1) + f_1(X_2)$ is independent with $e_1, e_2$, we can denote:

$$h(X_1 + X_2) := \hat{f}(X_1 + X_2, e) - f_1(e_1) - f_2(e_2) = f_1(X_1) + f_1(X_2) \tag{9}$$

Set $X_1 = X, X_2 = \mathbf{0}$, we have:

$$h(X) = f_1(X) + f_1(\mathbf{0}) = f_1(X)$$

Now we obtain $h \equiv f_1$. Thus, equation 9 imply:

$$f_1(X_1) + f_1(X_2) = f_1(X_1 + X_2)$$

for all $X_1$ and $X_2$.

Given that $f_1$ is differentiable, and by Cauchy's functional equation Kuczma (2009), it follows that $f_1(X) = AX$ for some matrix $A$. Thus, the function $f$ takes the form $f(X, e) = AX + f_2(e)$, concluding the proof. □

## B PROOFS FOR CONDITIONS OF CONDITIONAL INDEPENDENCE CONSISTENCY

**Theorem 3** (Necessary and Sufficient Condition for Conditional Independence Consistency). *Consider the distribution of $(S_X, S_Y, S_Z, Y_1, ..., Y_k)$ entailed from the aligned model, the following statements are equivalent when $2 \leq k < \infty$:*

    *(i) Conditional Independence:* $S_X \perp\!\!\!\perp S_Z \mid S_Y$

    *(ii) Conditional Probability Relation:* $\forall s_X, s_Y, s_Z \in \mathbb{R}$

$$\iint_{\mathbb{R}^k} \alpha(s_Z, s_Y, y_{1:k}) \left( \beta(y_{1:k}, s_Y, s_X) - \gamma(y_{1:k}, s_Y) \right) dy_1 ... dy_k = 0 \tag{10}$$

    *(iii) Alternative Conditional Probability Relation:* $\forall s_X, s_Y, s_Z \in \mathbb{R}$

$$\iint_{\mathbb{R}^k} \alpha^*(s_X, s_Y, y_{1:k}) \left( \beta^*(y_{1:k}, s_Y, s_Z) - \gamma(y_{1:k}, s_Y) \right) dy_1 ... dy_k = 0 \tag{11}$$

*where*

- $\alpha(s_Z, s_Y, y_{1:k}) := p_{S_Z|S_Y, Y_{1:k}}(s_Z|s_Y, y_{1:k})$

- $\beta(y_{1:k}, s_Y, s_X) := p_{Y_{1:k}|S_Y, S_X}(y_{1:k}|s_Y, s_X)$

- $\alpha^*(s_X, s_Y, y_{1:k}) := p_{S_X|S_Y, Y_{1:k}}(s_X|s_Y, y_{1:k})$

- $\beta^*(y_{1:k}, s_Y, s_X) := p_{Y_{1:k}|S_Y, S_X}(y_{1:k}|s_Y, s_X)$

- $\gamma(y_{1:k}, s_Y) := p_{Y_{1:k}|S_Y}(y_{1:k}|s_Y)$

*Proof.* The proof will proceed by showing that statements (i), (ii), and (iii) are mutually equivalent.

*Proof that (i) is equivalent to (ii):* Suppose that $S_X \perp\!\!\!\perp S_Z \mid S_Y$ holds. By the definition of conditional independence, this is equivalent to the statement that for all $s_X$, $s_Y$, and $s_Z$ in $\mathbb{R}$, we have $p_{S_Z|S_Y, S_X}(s_Z|s_Y, s_X) = p_{S_Z|S_Y}(s_Z|s_Y)$.

We can now derive both sides of this equation as follows.

On the left hand side(LHS):

$$p_{S_Z|S_Y,S_X}(s_Z|s_Y,s_X)$$
$$= \frac{p_{S_Z,S_Y,S_X}(s_Z,s_Y,s_X)}{p_{S_Y,S_X}(s_Y,s_X)}$$
$$= \frac{\iint_{\mathbb{R}^k} p_{S_Z|S_Y,S_X,Y_{1:k}}(s_Z|s_Y,s_X,y_{1:k})p_{S_Y,S_X,Y_{1:k}}(s_Y,s_X,y_{1:k})dy_1...dy_k}{p_{S_Y,S_X}(s_Y,s_X)}$$
$$= \iint_{\mathbb{R}^k} p_{S_Z|S_Y,Y_{1:k}}(s_Z|s_Y,y_{1:k})p_{Y_{1:k}|S_Y,S_X}(y_{1:k}|s_Y,s_X)dy_1...dy_k$$
$$= \iint_{\mathbb{R}^k} \alpha(s_Z,s_Y,y_{1:k})\beta(y_{1:k},s_Y,s_X)dy_1...dy_k$$

The first steps is based on the definition of conditional probability and the second step uses the law of total probability. The third step is using the d-separation: $\{Y_1,\ldots,Y_k\}$ d-separates $S_Z$ from $S_X$.

Meanwhile, RHS:

$$p_{S_Z|S_Y}(s_Z|s_Y)$$
$$= \frac{p_{S_Z,S_Y}(s_Z,s_Y)}{p_{S_Y}(s_Y)}$$
$$= \frac{\iint_{\mathbb{R}^k} p_{S_Z|S_Y,Y_{1:k}}(s_Z|s_Y,y_{1:k})p_{S_Y,Y_{1:k}}(s_Y,y_{1:k})dy_1...dy_k}{p_{S_Y}(s_Y)}$$
$$= \iint_{\mathbb{R}^k} \alpha(s_Z|s_Y,y_{1:k})\gamma(y_{1:k},s_Y)dy_1...dy_k$$

Finally, substitute both to the original equality:

$$p_{S_Z|S_Y,S_X}(s_Z|s_Y,s_X) - p_{S_Z|S_Y}(s_Z|s_Y)$$
$$= \iint_{\mathbb{R}^k} \alpha(s_X,s_Y,y_{1:k})\left(\beta(y_{1:k},s_Y,s_Z) - \gamma(y_{1:k},s_Y)\right)dy_1...dy_k$$
$$= 0$$

Hence, we arrive at the condition specified in (ii).

*Proof that (i) is equivalent to (iii)*: The proof that (i) and (iii) are equivalent is analogous to the above arguments. We therefore omit the details for brevity.

$\square$

**Corollary** (Sufficient Conditions for Conditional Independence). *If $\{S_X \perp\!\!\!\perp Y_{1:k} \mid S_Y\}$ or $\{S_Z \perp\!\!\!\perp Y_{1:k} \mid S_Y\}$ hold, then $S_X \perp\!\!\!\perp S_Z \mid S_Y$ holds.*

*Proof.* This corollary introduces two sufficient conditions for conditional independence. While the proofs for each are analogous, we demonstrate the proof for the first condition to avoid redundancy.

*Proof that $\{S_X \perp\!\!\!\perp Y_{1:k} \mid S_Y\}$ implies $S_X \perp\!\!\!\perp S_Z \mid S_Y$*:

Assume that $\{S_X \perp\!\!\!\perp Y_{1:k} \mid S_Y\}$ is true. By definition, this is equivalent to

$$p_{Y_{1:k}|S_Y,S_X}(y_{1:k}|s_Y,s_X) = p_{Y_{1:k}|S_Y}(y_{1:k}|s_Y)$$

for all $s_X$, $y_{1:k}$, and $s_Y$. Utilizing the notation from Theorem 2, we can rewrite this as

$$\beta(y_{1:k},s_Y,s_X) = \gamma(y_{1:k},s_Y).$$

This simplification makes it clear that the equality conforms to the second condition of Theorem 2, which is

$$\iint_{\mathbb{R}^k} \alpha(s_Z, s_Y, y_{1:k}) \left(\beta(y_{1:k}, s_Y, s_X) - \gamma(y_{1:k}, s_Y)\right) dy_1 \ldots dy_k = 0.$$

Because this holds for all $s_X$, $s_Y$, and $s_Z$ in $\mathbb{R}$, it follows that $S_X \perp\!\!\!\perp S_Z \mid S_Y$ is true, completing our proof.

$\square$

**Corollary.**   *1. For a fork-like aligned model:*

- *If $f_Z(Y_t, Z_{t-1}, e_{Z,t})$ is of the form $\alpha * Y_t + e_t$, where $\alpha$ can be any real number, then $S_Z \perp\!\!\!\perp Y_{1:k} \mid S_Y$.*
- *If $f_X(X_{t-1}, Y_t, e_{X,t})$ is of the form $\alpha * Y_t + e_t$, where $\alpha$ can be any real number, then $S_X \perp\!\!\!\perp Y_{1:k} \mid S_Y$.*

*2. For a chain-like aligned model:*

- *If $f_Z(Y_t, Z_{t-1}, e_{Z,t})$ is of the form $\alpha * Y_t + e_t$, where $\alpha$ can be any real number, then $S_Z \perp\!\!\!\perp Y_{1:k} \mid S_Y$.*
- *If the time series is stationary and Gaussian, and $f_Y(X_t, Y_{t-1}, e_{Y,t})$ is of the form $\alpha * X_t + e_t$, where $\alpha$ can be any real number, then $S_X \perp\!\!\!\perp Y_{1:k} \mid S_Y$.*

*Proof.* The proof for these four sufficient conditions is tied to the bivariate substructure within the fork and chain models.

We categorize the bivariate substructures within these trivariate structures into two types. The first type is where the middle node directs the side nodes, such as in the fork structure where the middle node $Y$ directs $X$ and $Z$. There are two such substructures in the fork model and one in the chain model where $Y$ directs $Z$. The second type is where the side node directs the middle node, seen in the chain model with $X$ directing $Y$.

Due to the causal direction in the bivariate structure, the sufficient conditions for $S_Z \perp\!\!\!\perp Y_{1:k} \mid S_Y$ and $S_X \perp\!\!\!\perp Y_{1:k} \mid S_Y$ in the fork, and $S_Z \perp\!\!\!\perp Y_{1:k} \mid S_Y$ in the chain are similar and share a similar proof. The sufficient condition for $S_X \perp\!\!\!\perp Y_{1:k} \mid S_Y$ in the chain is different from the other three. To avoid redundancy, we provide a proof for the sufficient condition for $S_Z \perp\!\!\!\perp Y_{1:k} \mid S_Y$ in the chain; the proof for the two conditions in the fork model is similar to this. We also provide the proof for the sufficient condition for $S_X \perp\!\!\!\perp Y_{1:k} \mid S_Y$ in the chain.

*proof for sufficient condition of $S_Z \perp\!\!\!\perp Y_{1:k} \mid S_Y$ in chain model:*

Suppose $f_Z(Y_t, e_{Z,t}) = \alpha * Y_t + e_{Z,t}$ for some real number $alpha$. Then, by substitution, we get

$$S_Z = \sum_{t=1}^{k} Z_t$$
$$= \sum_{t=1}^{k} (\alpha Y_t + e_{Z,t})$$
$$= \alpha S_Y + \sum_{t=1}^{k} e_{Z,t}$$

Given $S_Y$, the random part of $S_Z$ is only $\sum_{t=1}^{k} e_{Z,t}$, which is independent of $Y_{1:k}$. Therefore, it follows that $S_Z \perp\!\!\!\perp Y_{1:k} \mid S_Y$.

*proof for sufficient condition of $S_X \perp\!\!\!\perp Y_{1:k} \mid S_Y$ in chain model:*

We will prove the case for $k = 2$, and it can be easily generalized to $k \geq 3$. In the linear, Gaussian, stationary model for $k = 2$, we have:

$$X_1 \sim \mathcal{N}(0, \sigma_{X_1}^2)$$
$$Y_1 = \alpha X_1 + e_{Y,1}$$
$$X_2 = \beta X_1 + e_{X,2}$$
$$Y_2 = \alpha X_2 + e_{Y,2}$$

where $e_{X,2} \sim \mathcal{N}(0, \sigma_{e_X}^2)$, $e_{Y,1}, e_{Y,2}$ i.i.d. $\sim \mathcal{N}(0, \sigma_{e_Y}^2)$. And due to stationarity, $\sigma_{X_1}^2 = \frac{\sigma_{e_X}^2}{1 - \beta^2}$.

In the linear Gaussian case, conditional independence implies that the partial correlation equals 0. We have:

$$\text{cov}_{S_X, Y_1 | S_Y} = \text{cov}_{S_X, Y_1} - \frac{\text{cov}_{S_X, S_Y} \text{cov}_{Y_1, S_Y}}{\text{var}_{S_Y}}$$
$$\text{cov}_{S_X, Y_2 | S_Y} = \text{cov}_{S_X, Y_2} - \frac{\text{cov}_{S_X, S_Y} \text{cov}_{Y_2, S_Y}}{\text{var}_{S_Y}}$$

where

$$\text{cov}(S_X, Y_1) = \text{cov}(X_1, Y_1) + \text{cov}(X_2, Y_1),$$
$$\text{cov}(S_X, Y_2) = \text{cov}(X_1, Y_2) + \text{cov}(X_2, Y_2),$$
$$\text{cov}(S_X, S_Y) = \text{cov}(S_X, Y_1) + \text{cov}(S_X, Y_2),$$
$$\text{cov}(Y_1, S_Y) = \text{var}(Y_1) + \text{cov}(Y_1, Y_2),$$
$$\text{cov}(Y_2, S_Y) = \text{var}(Y_2) + \text{cov}(Y_1, Y_2).$$

Substitute these into the partial covariance equations to get

$$\text{cov}_{S_X, Y_1 \cdot Z} = \text{cov}_{S_X, Y_2 \cdot Z} = 0$$

$\square$

## C  CAUSAL DISCOVERY FROM AGGREGATED DATA

To investigate the direct effects of aggregation on causal discovery, we applied three widely-used causal discovery methods on both the original and aggregated data, comparing the results in both linear and nonlinear scenarios. The performance of these methods is measured using the correct rate over 100 repetitions.

For data generation, in each repetition, the original data is generated based on the causal graph shown in Figure 4. The causal relationships are defined as:

$$Z = X + Y + e_Z,$$
$$H = Z + e_H \quad \text{(for linear)};$$
$$Z = X^2 + Y^2 + e_Z,$$
$$H = Z^2 + e_H \quad \text{(for nonlinear)}.$$

The aggregated data is the result of aggregation with a factor of $k = 2$ based on the aligned model 2, having an instant structure resembling the original data. All datasets have a sample size of 500.

Regarding the method parameters, we used the Fisher-Z test for PC and FCI, and the BIC score for GES in the linear scenario. In the nonlinear scenario, we set the conditional independence test for PC and FCI as the Kernel Conditional Independence Test (KCI) with the default kernel and chose the "local score CV general" score function for GES. All other settings are kept at their default values.

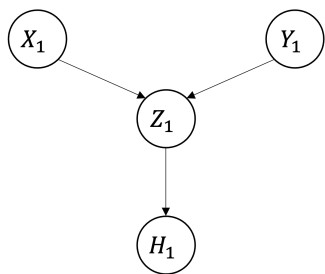

Figure 4: Causal graph of original data

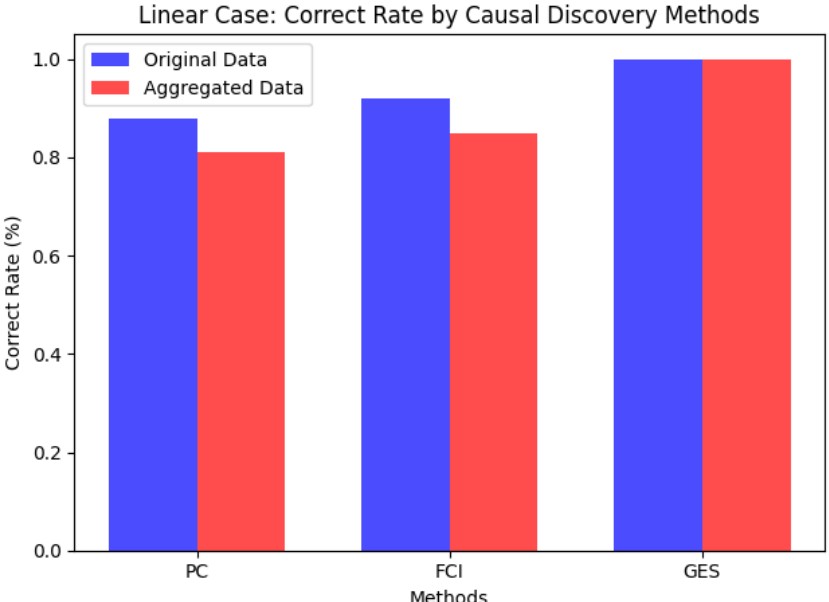

Figure 5: Linear Case: Correction Rate by Causal Discovery Method

From Figure 5, we observe that, in the linear scenario, aggregation does not adversely affect the performance of the causal discovery methods. This might explain why the causal community has not prioritized the aggregation issue in instantaneous causal discovery for a long time.

Contrastingly, the nonlinear scenario paints a completely different picture, with aggregation causing a significant drop in the performance of all three methods. It is crucial to rigorously investigate this issue to understand its causes and potential solutions.

## D EXPERIMENT FOR FUNCTIONAL CONSISTENCY

Determining the causal direction between two variables is an essential task in causal discovery. To understand the impact of aggregation on this task, we employed two renowned FCM-based causal discovery methods: Direct LiNGAM for the linear scenario and Additive Nonlinear Model (ANM) for the nonlinear one. We assessed how the correct rate in 100 repetitions varies with the aggregation factor $k$.

For data generation, it's straightforward. The data is generated based on an aligned model with an instantaneous causal relationship, given by:

$$Y = 2X + e_Y \quad \text{(for linear)};$$
$$Y = X^2 + e_Y \quad \text{(for nonlinear)},$$

where the independent noise follows a standard uniform distribution. The sample size is 500.

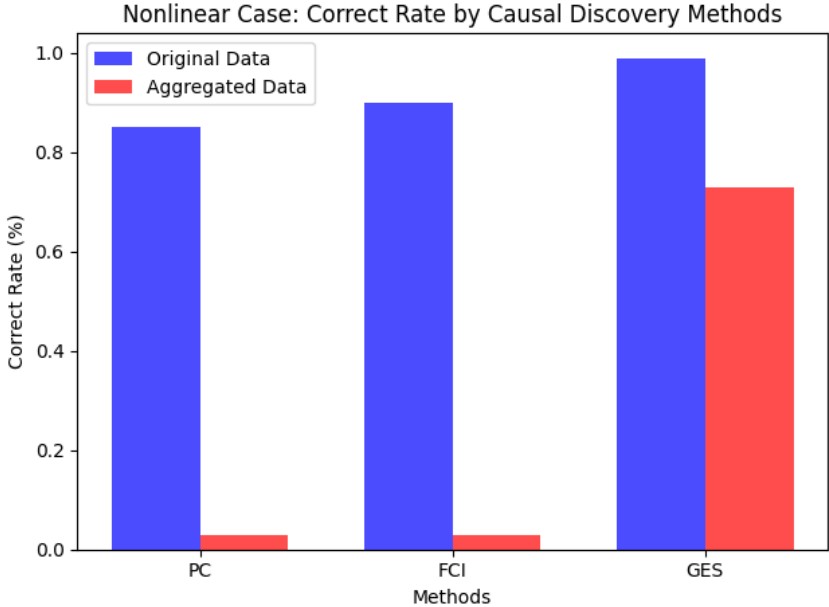

Figure 6: Nonlinear Case: Correction Rate by Causal Discovery Method

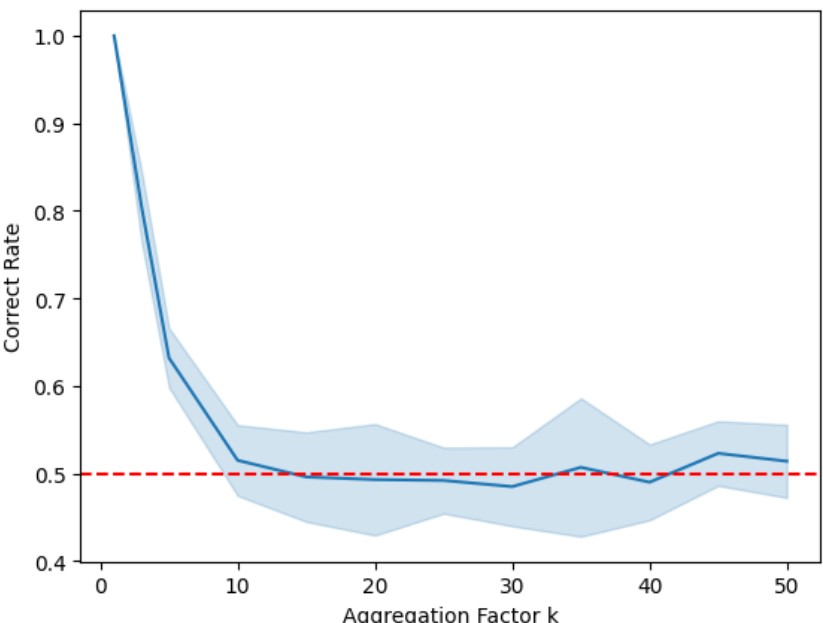

Figure 7: Linear Case: Direct LiNGAM Correction Rate with Different Aggregation Factors $k$

From the presented figures, it's evident that in the linear non-Gaussian case, the non-Gaussian distribution increasingly approaches a Gaussian one as $k$ grows. Eventually, Direct LiNGAM resembles a random guess. It's noteworthy that the x-axis range for the linear scenario spans from 0 to 50, while for the nonlinear case, it's from 0 to 10. This difference suggests that the performance of ANM deteriorates faster than Direct LiNGAM. ANM is not reliant on non-Gaussian properties but on additive noise. Aggregated data lacks functional consistency as the additive noise function is disrupted by the aggregation, rendering ANM ineffective.

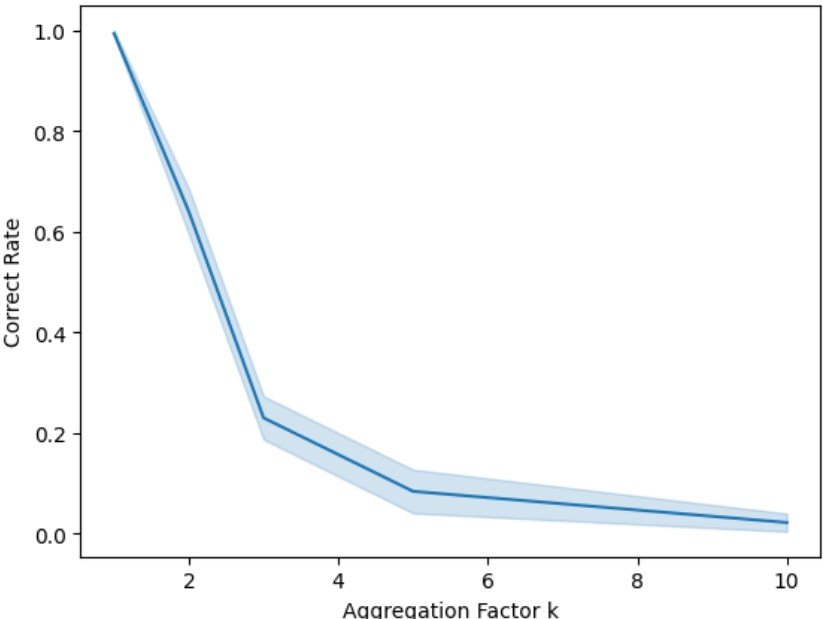

Figure 8: Nonlinear Case: ANM Correction Rate with Different Aggregation Factors $k$

## E    EXPERIMENT FOR CONDITIONAL INDEPENDENCE CONSISTENCY

We conduct experiments using three different structures: chain, fork, and collider, to validate our theoretical results in Section 4. However, due to space constraints, we only report the results for the fork-like model in the main text. In this section, we will reiterate the experiment details and report the complete results for chain, fork, and collider structures, along with a more detailed analysis.

The specific settings for these structures are as follows:

**Chain-like Model:**  $X_t = e_{X,t} , Y_t = f(X_t, e_{Y,t}) , Z_t = g(Y_t, e_{Z,t})$
**Fork-like Model:**  $X_t = f(Y_t, e_{X,t}), Y_t = e_{Y,t}, Z_t = g(Y_t, e_{Z,t})$
**Collider-like Model:**  $X_t = e_{X,t} , Y_t = f(X_t, e_{Y,t}) + g(Z_t, e_{Y,t}) , Z_t = e_{Z,t}$

In all structures, each noise term $e$ follows an independent and identically distributed (i.i.d) standard Gaussian distribution. For the causal relationship between $X$ and $Y$, we use the function $f(\cdot, e)$. In the linear case, $f(\cdot) = (\cdot)$. In the nonlinear case, we use the post-nonlinear model $f(\cdot, e) = G(F(\cdot) + e)$ Zhang & Hyvarinen (2012) and uniformly randomly pick $F$ and $G$ from $(\cdot)^2$, $(\cdot)^3$, and $tanh(\cdot)$ for each repetition. This is to ensure that our experiment covers a wide range of nonlinear cases. Similarly, the same approach is applied for the relationship between $Y$ and $Z$ with the corresponding function $g(\cdot)$.

And t=1,2, $S_X = X_1 + X_2, S_Y = Y_1 + Y_2, S_Z = Z_1 + Z_2$. And we generate 1000 i.i.d. data points for $(X_1, X_2, Y_1, Y_2, S_X, S_Y)$ and feed them into the approximate kernel-based conditional independence test Strobl et al. (2019). We test the null hypothesis(conditional independence) for (I) $S_X \perp\!\!\!\perp S_Y$, (II) $S_Y \perp\!\!\!\perp S_Z$, (III) $S_X \perp\!\!\!\perp S_Z$, (IV) $S_X \perp\!\!\!\perp S_Y \mid S_Z$, (V)$S_Y \perp\!\!\!\perp S_Z \mid S_X$, (VI)$S_X \perp\!\!\!\perp S_Z \mid S_Y$. And we also test for the conditional independence in corollary 1: (A)$S_X \perp\!\!\!\perp Y_1 \mid S_Y$, and (B)$S_Z \perp\!\!\!\perp Y_1 \mid S_Y$. We report the rejection rate, rounded to the nearest percent, at a 5% significance level in 1000 repeated experiments in Table 2a, 2b, 2c.

This experiment support our theoretical results, suggesting that conditional independence consistency can be ensured even with some nonlinearity in the model.

Specifically, let's examine the results for chain and fork. We anticipate the tested conditional independence set to contain only $S_X \perp\!\!\!\perp S_Z \mid S_Y$. If so, we can assert that temporal aggregation maintains conditional independence consistency.

Table 2: Rejection rates for CIT tests with different combinations of linear and nonlinear relationships. The index in the box represents the conditional independence that the structure should ideally have. Simply speaking, for the column with the index in the box, the closer the rejection rate is to 5%, the better. For all other columns from I to VI, a higher rate is better.

(a) Chain Structure

| $X_t \to Y_t$ | $Y_t \to Z_t$ | I | II | III | IV | V | VI | A | B |
|---|---|---|---|---|---|---|---|---|---|
| Linear | Linear | 100% | 100% | 100% | 100% | 100% | 6% | 4% | 7% |
| Nonlinear | Linear | 92% | 100% | 89% | 63% | 100% | 9% | 27% | 10% |
| Linear | Nonlinear | 100% | 95% | 87% | 100% | 94% | 5% | 6% | 82% |
| Nonlinear | Nonlinear | 92% | 86% | 56% | 89% | 85% | 18% | 27% | 39% |

(b) Fork Structure

| $X_t \to Y_t$ | $Y_t \to Z_t$ | I | II | III | IV | V | VI | A | B |
|---|---|---|---|---|---|---|---|---|---|
| Linear | Linear | 100% | 100% | 100% | 100% | 100% | 5% | 6% | 5% |
| Nonlinear | Linear | 92% | 100% | 84% | 92% | 100% | 5% | 76% | 5% |
| Linear | Nonlinear | 100% | 93% | 85% | 100% | 93% | 5% | 5% | 71% |
| Nonlinear | Nonlinear | 92% | 93% | 72% | 86% | 87% | 58% | 72% | 74% |

(c) Collider Structure

| $X_t \to Y_t$ | $Y_t \to Z_t$ | I | II | III | IV | V | VI | A | B |
|---|---|---|---|---|---|---|---|---|---|
| Linear | Linear | 100% | 100% | 5% | 100% | 100% | 99% | 4% | 5% |
| Nonlinear | Linear | 95% | 89% | 5% | 96% | 91% | 51% | 17% | 56% |
| Linear | Nonlinear | 90% | 95% | 5% | 91% | 96% | 48% | 54% | 15% |
| Nonlinear | Nonlinear | 81% | 81% | 6% | 83% | 81% | 29% | 26% | 26% |

From the first and second tables for chain and fork, it's evident that when the model is entirely nonlinear, the results for conditional independence can be erroneous. For instance, the rejection rate for conditional independence that should have been rejected is not high. In the chain structure, the rejection rate for the conditional independence III is zero, implying that every conditional independence test wrongly accepted this conditional independence (type II error). Conversely, the conditional independence that should have been accepted, VI $S_X \perp\!\!\!\perp S_Z \mid S_Y$, has rejection rates of 62% (chain) and 36% (fork), significantly exceeding the significance level of 5%. This aligns with our conclusion in remark 1, stating that chain and fork models cannot guarantee conditional independence consistency in general cases.

However, when half the model is linear, all conditional independence that should be rejected exhibit high rejection rates, indicating fewer type II errors. Moreover, the rejection rate for the acceptable conditional independence VI is quite low, closely approximating the significance level of 5%. This suggests that conditional independence-based causal discovery methods can still be applied to temporally aggregated data when the system is partially linear.

Columns A and B primarily aim to validate corollary 2 and corollary 1. The conditional independence represented by A and B corresponds to the two sufficient conditions for conditional independence consistency in corollary 1. From the experimental results, we find that if one of these conditions holds, we can ensure conditional independence consistency. For example, in the fork results, under the nonlinear+linear case, B holds while A does not. Nonetheless, we still have conditional independence consistency. Moreover, our findings further verify corollary 2, indicating that when a certain part of the causal mechanism is linear, the corresponding sufficient condition in this part is satisfied, ultimately ensuring the conditional independence consistency of the entire system.

Finally, looking at the collider results, conditional independence consistency is maintained under all nonlinear and linear combinations, which agrees with our conclusion in remark 1, stating that collider can ensure conditional independence consistency under general conditions.

## F    EFFECT OF $k$ VALUE

Gong et al. (2017) have proven that the time-delay causal model (time series data) will transform into an instantaneous causal model (i.i.d. data). In our paper, we utilize large $k$ values to approximate the aggregation of the time-delay model as the aggregation of the instantaneous model (aligned model as defined in the main text). We aim to demonstrate the reasonableness of this approximation and to show how quickly the time-delay model transitions to an instantaneous model.

We apply the GES method on a linear fork-like time-delay model (VAR) and a fork-like instantaneous model (aligned model) across different values of $k$.

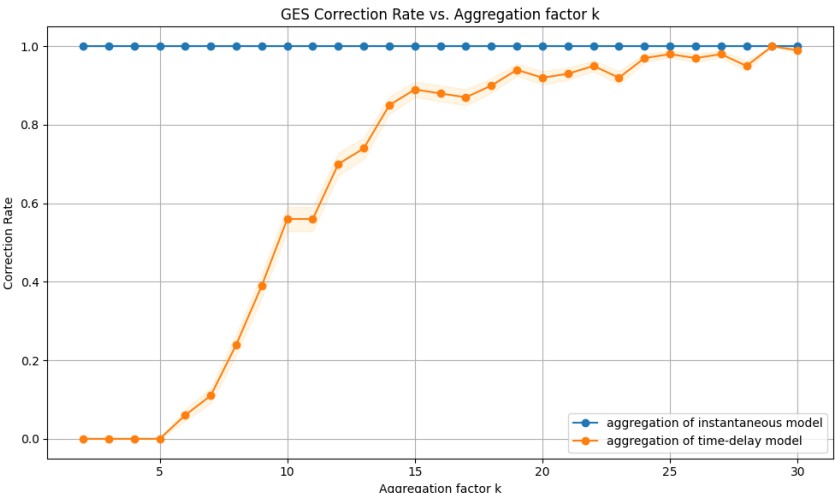

Figure 9: GES Correct Rate vs. Factor $k$

From Figure 9, we observe that $k$ does not need to be infinite. A sufficiently large $k$, such as 20, ensures that the time-delay model becomes an instantaneous model detectable by the instantaneous causal discovery method.

## G    CAUSAL DISCOVERY ON AGGREGATED DATA WITH PRIOR KNOWLEDGE

Inspired by Remark 1, we propose a straightforward solution to ensure that the PC algorithm identifies the correct Markov equivalence. The remark indicates that the collider structure retains conditional independence consistency. This implies that the PC algorithm can determine the correct v-structure if it has the correct skeleton. We therefore conducted experiments to assess whether the PC with a given skeleton as prior knowledge can discern the correct Markov equivalence.

From Figure 10, it becomes evident that when the PC is applied directly to aggregated data, its performance is subpar. However, when provided with the skeleton as prior knowledge, the PC consistently identifies the correct v-structure, yielding accurate results(see Figure 11). This discovery suggests that future work should concentrate on addressing the aggregation problem during the skeleton discovery process.

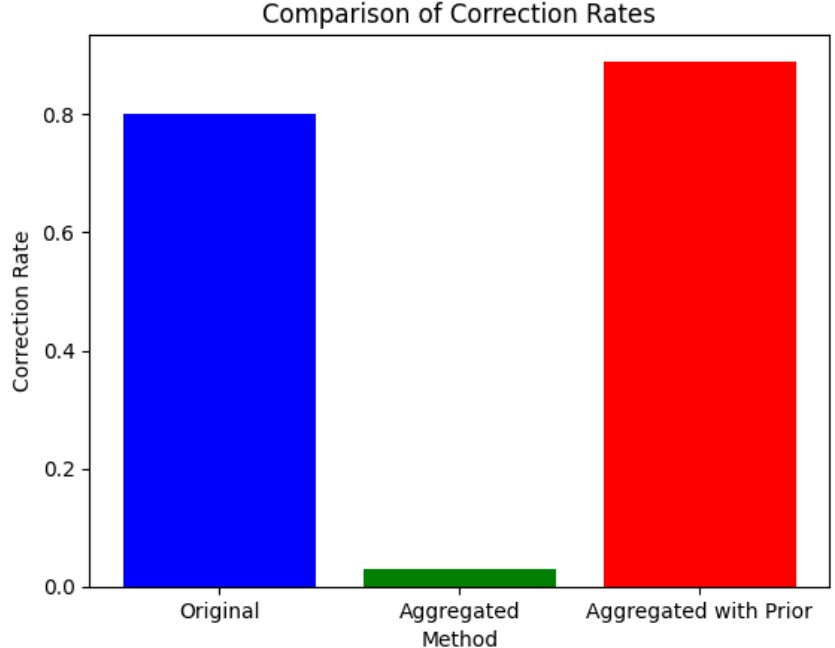

Figure 10: Correct Rate: PC on Original Data vs. PC on Aggregated Data vs. PC with Prior on Aggregated Data

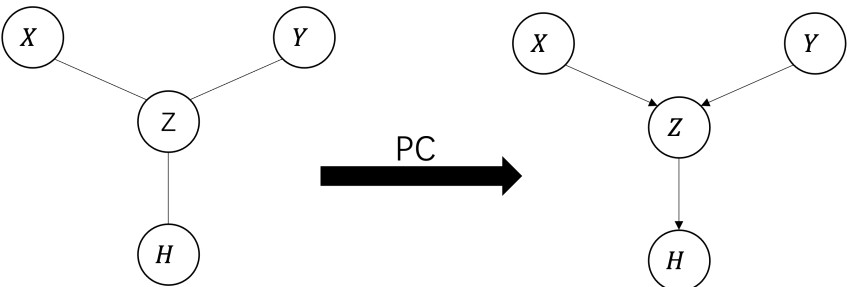

Figure 11: PC can find correct v-structure on aggregated data

