# OpenReview forum: "On the Recoverability of Causal Relations from Temporally Aggregated I.I.D Data"
_ICLR.cc/2024/Conference — Submitted to ICLR 2024_

### Official Review · Reviewer_yyms · 2023-10-30

**Soundness:** 3 good
**Presentation:** 2 fair
**Contribution:** 3 good
**Rating:** 6
**Confidence:** 3

**Summary:**

This paper studies the problem of Causal Discovery for Time-Series Data with IID samples. This problem is relevant for a variety of applications, such as healthcare, bioinformatics, economics and finance. In this submission, the author(s) study if causal discovery results obtained from IID time series data is consistent with the true causal process. To this end, the author(s) provide various theoretical results and experiments. The author(s) conclude that functional consistency is difficult to achieve in non-linear situations, and that even in linear non-Gaussian situations, the instantaneous model generated by temporal aggregation is still unidentifiable as the number the number of data points from the underlying causal process that are combined to form each observation goes to infinity.

**Strengths:**

In my opinion, this paper has the following strengths: (i) The problem is overall very relevant; (ii) the paper is well-written and easy to follow; (iii) the theoretical results are interesting. Overall, I believe that a discussion on identifiability and functional consistency of time-series data is needed.

**Weaknesses:**

Although this paper is mostly of theoretical interest, it would have been good to have more experiments. In the main text, the author(s) only consider simulated experiments on conditional independence tests and LiNGAM.

**Questions:**

In your submission, you discuss the relationship with faithfulness and conditional independence consistency. However, it can be argued that faithfulness is generally an unrealistic assumption in practice. Have you consider weaker assumptions, such as Causal Minimality (see i.e. Assumption 3 in [1])?

[1] https://arxiv.org/abs/2210.14706

---

> ### Author Response · Authors · 2023-11-20
>
> We are very grateful for the constructive comments and the time devoted to our work.
>
> > [W1] Although this paper is mostly of theoretical interest, it would have been good to have more experiments. In the main text, the author(s) only consider simulated experiments on conditional independence tests and LiNGAM.
>
>
> Due to the page limit, most of the pages are dedicated to presenting theoretical results, leading to only a limited number of experimental results included in the main text. The methods we have tested encompass PC, FCI, GES, LiNGAM, and ANM. The settings cover different aggregation levels and scenarios, including linear/non-linear scenarios, with and without prior knowledge, and the aggregation of both time-delay and instantaneous models.
>
> We have included 5 experiments in our original submission:
>
> * applied widely-used causal discovery methods to aggregation data(Appendix C)
> * functional consistency(Appendix D)
> * conditional independence consistency(Appendix E)
> * the impact of the k value(Appendix F)
> * preliminary solution to the aggregation problem(Appendix G)
>
>
> > [Q1] In your submission, you discuss the relationship with faithfulness and conditional independence consistency. However, it can be argued that faithfulness is generally an unrealistic assumption in practice. Have you consider weaker assumptions, such as Causal Minimality (see i.e. Assumption 3 in [1])?
>
> We completely agree that 'faithfulness is generally an unrealistic assumption in practice.' In fact, in our paper, we only use the faithfulness assumption as a starting point in Remark 1. The remainder of that section is dedicated to discussing how violating the faithfulness assumption can achieve conditional independence consistency.

---

> ### Author Response · Authors · 2023-11-22
> **Look Forward to Your Feedback**
>
> Thank you once again for the time you've dedicated to reviewing our paper. The discussion period will end soon, and we would appreciate the reviewers' feedback. Specifically, if the reviewers have further questions or concerns after reading the rebuttal, we hope to have the opportunity to address them.

---

### Official Review · Reviewer_2HMB · 2023-10-31

**Soundness:** 3 good
**Presentation:** 3 good
**Contribution:** 2 fair
**Rating:** 6
**Confidence:** 4

**Summary:**

The authors focus on the problem of temporal aggregation.  In many real-world problems, data is aggregated at a very high scale, which may lead highly-dependent data to become i.i.d.  If we're interested in effects at the de-aggregated level, performing causal discovery at the aggregate level can lead to errors.  The authors define conditions for functional and conditional independence consistency under temporal aggregation.  They then provide some simulation experiments to demonstrate the dangers of aggregation.

**Strengths:**

The authors address a very important problem - they are correct that temporal aggregation is rampant and, if done carelessly, can result in problems such as the introduction of spurious effects.  For the most part, the authors motivate this well.  The division into functional consistency and CI consistency is logical and well-presented and, for the most part, the notation is clear.  The experiments that are there do a good job at providing an initial assessment of the soundness of their findings.

**Weaknesses:**

I think much of this paper is poorly explained, leading to some issues with understandability.  There are a few sentences, that occur at unfortunately important parts, that I had a hard time following.  I especially noted this in the description of the intuition behind Definition 1 and Theorem 1.
- For the paragraph after Definition 1, that first sentence is a run-on ("Intuitively, functional consistency might suggest X=f(X,e), here we introduce a different function f, which allows for the possibility"), and the sentence after that sounds very strange ("This is because actually existing such f is difficult for nonlinear case").
- Theorem 1 feels impossible to follow grammatically. "f has the form ___ is necessary for the functional equation ____ holds for some f and some e that is only related to e with ______"
Theorem 2 could also benefit from some discussion around intuition.  My understanding is that Theorems 1 and 2 are the main results of this paper (as the conditions for consistency for functional and CI-based methods).  However, Theorem 1 suffers from the grammatical lac of clarity I described above, and Theorem 2 introduces a whole new set of variables (alpha, beta, and gamma) that don't appear to ever be defined (though please let me know if I just missed them and they are defined somewhere).  Some intuitive discussion of these theorems, as well as an editing pass to increase clarity, would improve this paper significantly.

In Section 3.1, after equation 1, you suggest that g(k) can be any normalizing function, such as g(k)=1 or g(k)=k.  However, later in that section, you say that, as k becomes large, g(k) approaches 0, allowing you to remove that term under large k.  However, I'm not sure how that works in general when you say that g(k)=1 is a valid normalizing function (which will not tend to 0).  And g(k)=k will only tend to 0 with very large k, but no discussion (that I noticed) is ever had about how large k needs to be for the results of this paper to hold.  Especially since in the experiments, the largest k I see is 50, which doesn't feel large enough to assume that g(k) -> 0.

While the authors discuss the dangers of temporal aggregation at a high level in the intro, the only example actually provided in the intro (the effect of temperature on ice cream sales, aggregated daily) is actually one where temporal aggregation doesn't seem to pose a problem.  While the effects become instantaneous at the aggregate level, it's an effect that's consistent with the dynamics at the de-aggregated level and one that seems reasonable for causal discovery methods to detect.  With the focus on dangers of potentially spurious instantaneous relationships from aggregation, an example where such relationships can arise would be helpful for motivation.

The experiments are quite weak.  The authors present two sets of experiments: a simulation with a single, two-variable model that shows how Direct LiNGAM has issues as k increases (due to the non-Gaussianity assumpton?) and experiments with a three variable model with either a chain, fork, or collider structure, showing how CI tests perform poorly if no linearity is present.  If I'm understanding correctly, the second set of experiments doesn't actually contain any aggregation (it takes place over two time steps) and doesn't actually employ a specific causal discovery algorithm, focusing instead of the performance of CI tests.  These are reasonable first-step tests, but the lack of any remotely realistic data, data with more than 2-3 variables or across more than 2 time steps, and structures with any amount of complexity leaves the evaluation feeling very weak.

**Questions:**

In Section 4, why are only chain-like, fork-like, and collider-like models considered?  Given that the focus is on temporal aggregation, cycle structures are quite likely (e.g., X{t-1} -> Y{t}, Y{t-1} -> Z{t}, Z{t-1} -> X{t}).  Obviously, this causes problems upon aggregation if we don't have a way for our model to learn or reason about cyclic models, but this seems like the type of case that is especially relevant in a discussion about temporal aggregation.

---

> ### Author Response · Authors · 2023-11-20
> **Official Comment by Authors(1/2)**
>
> We are very grateful for the constructive comments and the time devoted to our work.
>
> > [W1] ...I had a hard time following. I especially noted this in the description of the intuition behind Definition 1 and Theorem 1.
>
> Sorry to cause vagueness. We modified Definition 1 and Theorem 1 and rewrite the explanation for Definition 1 in revised version.
>
>
>
> Definition 1:
>
> Functional Consistency means that there exists a function $\hat{f}$ such that the temporally aggregated data \(\overline{X}\) is described by $\overline{X} = \hat{f}(\overline{X}, e)$, where $e$ is a noise vector with independent components that are only related to the independent noises encountered in the process, denoted as $e_{2:k+1}$.
>
> Explanation:
>
> This definition implies that if functional consistency holds, then the aggregated data can at least be represented as a Structural Causal Model (SCM) in vector form, and the source of independent noise aligns with the underlying process. Please note that we allow the generative mechanism $\hat{f}$ to differ from the underlying causal function $f$. However, even with this allowance, achieving functional consistency in the nonlinear case remains challenging, as we will demonstrate in this section.
>
> Theorem 1:
>
> Consider a function $f(X, e)$ that is differentiable with respect to $X$. Define the following:
>
> Statement 1: The function $f$ is of the form $f(x, e) = Ax + f_2(e)$ for some function $f_2$.
>
> Statement 2: For any positive integer $k$, there exists a function $\hat{f}$ such that the functional equation
> $\frac{\sum_{i=1}^k f(X_{i}, e_{i+1})}{g(k)} = \hat{f}(\overline{X}, e)$
> holds for any $X_i$, $e_i$, and any normalization factor $g(k)$, where $e$ is related only to $e_i$ for $i = 2, \dots, k+1$.
>
> Statement 1 is a necessary condition for Statement 2.
>
> > Theorem 2 introduces a whole new set of variables (alpha, beta, and gamma) that don't appear to ever be defined
>
> The definitions of the notations alpha, beta, and gamma have been provided within the content of Theorem 2 in the original manuscript.
>
>
>
> > [W2] In Section 3.1, after equation 1, you suggest that g(k) can be any normalizing function, such as g(k)=1 or g(k)=k ... but no discussion (that I noticed) is ever had about how large k needs to be for the results of this paper to hold. Especially since in the experiments, the largest k I see is 50, which doesn't feel large enough to assume that g(k) -> 0.
>
>
>
> We have corrected this in the revised version. We do need $\lim_{k \to \infty} g(k) = +\infty$ for the approximation part. However, when our analysis begins with the aggregation of the instantaneous model and considers any finite $k$, the choice of $g(k)$ does not matter at all.
>
> For how large $k$ needs to be, please note that in our paper, we only require a large $k$ to diminish unaligned terms for approximation/alignment. We need to approximate the aggregation of the time-delay model as the aggregation of the instantaneous model. Once we start the analysis from an aligned model, the value of $k$ is no longer a concern. $k$ can be any finite integer larger or equal to 2. In the experiment in Appendix F, we discuss how large a $k$ value is sufficient for approximation. The experiments show that when $k = 30$, the results of the aggregation of the time-delay model and the aggregation of the instantaneous model are very similar, which is sufficient to make the approximation reasonable.
>
>
>
>
>
> > [W3] While the authors discuss the dangers of temporal aggregation at a high level in the intro, the only example actually provided in the intro (the effect of temperature on ice cream sales, aggregated daily) is actually one where temporal aggregation doesn't seem to pose a problem. While the effects become instantaneous at the aggregate level, it's an effect that's consistent with the dynamics at the de-aggregated level and one that seems reasonable for causal discovery methods to detect...
>
>
> “While the effects become instantaneous at the aggregate level, it's an effect that's consistent with the dynamics at the de-aggregated level and one that seems reasonable for causal discovery methods to detect.” No. The aggregation of an instantaneous causal model still suffers from the distortions caused by aggregation.
>
> All theoretical results in our paper are about the aggregation of instantaneous effects. The damage to causal discovery performance is due not only to time-delay causal models but also to the aggregation itself. The primary focus of our paper is to investigate the effects caused by aggregation itself, underscoring how it can potentially distort causal relationships.

---

> ### Author Response · Authors · 2023-11-20
> **Official Comment by Authors(2/2)**
>
> > [W4] The experiments are quite weak...but the lack of any remotely realistic data, data with more than 2-3 variables or across more than 2 time steps, and structures with any amount of complexity leaves the evaluation feeling very weak.
>
> Due to the page limit, most of the pages are dedicated to presenting theoretical results, leading to only a limited number of experimental results included in the main text. The methods we have tested encompass PC, FCI, GES, LiNGAM, and ANM. The settings cover different aggregation levels and scenarios, including linear/non-linear scenarios, with and without prior knowledge, and the aggregation of both time-delay and instantaneous models.
>
> We have included 5 experiments in our original submission:
>
> * applied widely-used causal discovery methods to aggregation data(Appendix C)
> * functional consistency(Appendix D)
> * conditional independence consistency(Appendix E)
> * the impact of the k value(Appendix F)
> * preliminary solution to the aggregation problem(Appendix G)
>
> "doesn't actually employ a specific causal discovery algorithm": Please see Appendix C: we applied widely-used causal discovery methods(PC, FCI, GES) to aggregation data with 4 variables.
>
> "data with more than 2-3 variables": Please see Appendix C,G
>
> "across more than 2 time steps": Please see Appendix D,F
>
> In the revised version, we have rewritten the first paragraph in the experiment section of the main text to clearly outline the experiments included, making it easier for readers to understand what experiments were conducted.
>
> For the statement 'the second set of experiments doesn't actually contain any aggregation.': If the data represents the aggregation of an instantaneous model, the experimental results for different aggregated levels k large or equal to 2 will not show much difference. Moreover, an aggregation level of k=2 still qualifies as aggregation.
>
> > [Q1] In Section 4, why are only chain-like, fork-like, and collider-like models considered? Given that the focus is on temporal aggregation, cycle structures are...
>
> We agree that the cyclic model is important in discussions about aggregation, and the analysis in Section 3 includes the cyclic model. However, Section 4, which focuses on conditional independence consistency, does not discuss the cyclic model. This is because, to our knowledge, the methods for discovering cyclic models are mainly FCM-based (discussed in Section 3) or score-based methods.
>
> Regarding your example, when structure is cyclic, there is no conditional independence involving $\overline{X}, \overline{Y}, \overline{Z}$. Aggregation generally makes conditional independence conditionally dependent; it does not change a dependent relationship into an independent one. Therefore, we believe that aggregation has no influence on conditional independence in your example.

---

> ### Author Response · Authors · 2023-11-22
> **Look Forward to Your Feedback**
>
> Thank you once again for the time you've dedicated to reviewing our paper. The discussion period will end soon, and we would appreciate the reviewers' feedback. Specifically, if the reviewers have further questions or concerns after reading the rebuttal, we hope to have the opportunity to address them.

---

> > ### Comment · Reviewer_2HMB · 2023-11-22
> >
> > I appreciate your detailed response.  The experiments in the appendix are very helpful, and I'm wondering if some of those are more compelling than the ones present in the main paper.  Particularly, the comparison of performance degradation for linear and nonlinear models (Figures 5 and 6) is very striking.  I'm also interested in the result presented in Appendix F, since the issue of "how big does k have to be?" seems to be a recurring one among the reviewers, due to the frequent mentions in the paper of assuming k is large.  I wish this had been explored more.  From Figure 9, the authors conclude "A sufficiently large k, such as 20,
> > ensures that the time-delay model becomes an instantaneous model detectable by the instantaneous
> > causal discovery method."  However, I'm not sure where the confidence that it "ENSURES that the time-delayed model becomes an instantaneous model [...]" is coming from.  Is it just from looking at Figure 9?  In that image, ~20 does seem to be where performance gets pretty similar, but it's unclear, from this single pair of experiments, how general that number is.
> >
> > As it stands, I'm willing to raise my score from a 5 to a 6.

---

> > > ### Author Response · Authors · 2023-11-23
> > >
> > > Thank you so much for your feedback and for updating the rating. We are very delighted that you kindly found our response helpful.
> > >
> > > > The experiments in the appendix are very helpful, and I'm wondering if some of those are more compelling than the ones present in the main paper. Particularly, the comparison of performance degradation for linear and nonlinear models (Figures 5 and 6) is very striking.
> > >
> > > Yes, Figures 5 and 6 effectively illustrate the motivation behind our paper, which is the observation that the performance of nonlinear causal discovery collapses in aggregation. However, we have presented one experiment to support the theoretical results in Section 3 and another to support Section 4. Due to page limitations, it is challenging to add an experiment specifically to illustrate this motivation.
> > >
> > >
> > >
> > > >  However, I'm not sure where the confidence that it "ENSURES that the time-delayed model becomes an instantaneous model [...]" is coming from. Is it just from looking at Figure 9? In that image, ~20 does seem to be where performance gets pretty similar, but it's unclear, from this single pair of experiments, how general that number is.
> > >
> > > We conducted another experiment to demonstrate the applicability of our theoretical results and the rate at which the approximation error diminishes as k increases. We performed conditional independence tests separately on the temporal aggregation of the time-delay model and the temporal aggregation of the instantaneous model with different values of k to test whether S_X is independent of $S_Z$ given $S_Y$.
> > >
> > > ### Table b: Rejection rate in 1000 repetitions
> > >
> > > #### k= 2
> > >
> > > |                | Rejection rate for data_ins | Rejection rate for data_delay |
> > > |----------------|-----------------------------|-------------------------------|
> > > | Linear+Linear  | 0.051                       | 1                             |
> > > | Nonlinear+Linear  | 0.059                       | 0.767                         |
> > > | Linear+Nonlinear  | 0.044                       | 0.767                         |
> > > | Nonlinear+Nonlinear  | 0.386                       | 0.772                         |
> > >
> > > #### k= 5
> > >
> > > |                | Rejection rate for data_ins | Rejection rate for data_delay |
> > > |----------------|-----------------------------|-------------------------------|
> > > | Linear+Linear  | 0.051                       | 1                             |
> > > | Nonlinear+Linear  | 0.056                       | 0.154                         |
> > > | Linear+Nonlinear  | 0.060                       | 0.151                         |
> > > | Nonlinear+Nonlinear  | 0.707                       | 0.778                         |
> > >
> > > #### k= 50
> > >
> > > |                | Rejection rate for data_ins | Rejection rate for data_delay |
> > > |----------------|-----------------------------|-------------------------------|
> > > | Linear+Linear  | 0.053                       | 0.122                         |
> > > | Nonlinear+Linear  | 0.046                       | 0.042                         |
> > > | Linear+Nonlinear  | 0.046                       | 0.047                         |
> > > | Nonlinear+Nonlinear  | 0.375                       | 0.382                         |
> > >
> > > From the table, when the original model is instantaneous, the conditional independence test results align with our theoretical findings for any value of k. This yields a rejection rate of around 0.05 for the first three cases and a higher rejection rate for the last case. This consistency is due to our theoretical results being applicable to the temporal aggregation of the instantaneous model for any finite k.
> > >
> > > For the time-delay original model, when k is extremely small, like 2, the experimental results don't align with our theoretical findings. This discrepancy arises because our theoretical results are tailored for larger finite values of k in this context.
> > >
> > > Furthermore, as k increases, the temporal aggregation of the time-delay model becomes increasingly similar to that of the instantaneous model. This implies that our approximation is reasonable for sufficiently large k. The data_delay also exhibits the expected rejection rate consistent with our theoretical results when k=50. This means that for a large enough value of k, our theoretical results are applicable to the temporal aggregation of the time-delay model.
> > >
> > > We acknowledge that the sufficient value of 'k' varies for different examples. In future versions, we plan to include a broader range of examples to discuss what constitutes a sufficient value for 'k' and edit the discussions according to the experimental results.

---

### Official Review · Reviewer_766L · 2023-10-31

**Soundness:** 2 fair
**Presentation:** 3 good
**Contribution:** 2 fair
**Rating:** 6
**Confidence:** 3

**Summary:**

The authors provide theoretical conditions that are necessary to ensure the consistency of causal discovery results, including functional consistency and conditional independence consistency when analyzing temporally aggregated i.i.d. data.

**Strengths:**

1. The problem the authors seek to address is an interesting question that holds significance in the field of causal discovery. The authors discussed conditions needed to recover true causal relations from temporally aggregated data.
2. The authors conduct simulation experiments to support their claims.

**Weaknesses:**

1. I doubt that there is a technical flaw in the proof of Theorem 3, which is described in the Questions section.
2. The credibility of Corollary 1 is also in question, as it follows Theorem 3. This is mentioned in the Questions section.
3. While the name of Theorem 3 is "Necessary and Sufficient Condition for Conditional Independence Consistency," the content of "Theorem 3" only presents three equivalent statements that appear to just utilize the marginalization of the joint distribution. Please briefly explain how the three equivalent statements are necessary and sufficient conditions for conditional independence consistency if I misunderstand the point.
4. As mentioned in the limitation section, the paper discussed the impact of temporal aggregation but did not offer any resolution. Consequently, it is challenging to assess the significance of this paper's contribution.
5. Given Fig.2 and the discussion between 4.2 and Remark 1, the experimental results appear to illustrate that discussion rather than provide evidence for any significant claims in the main paper.

**Questions:**

1. Referencing specific equations is challenging as there is no index for the equations in the appendix.
2. In the proof of Theorem 3, the derivation from the third equation and the fourth equation below the sentence "On the left hand side (LHS):" is in question. The denominator of the third equation is $p_{S_Y, S_X}(S_Y, S_X)$ and $S_Y$ is a function of $Y_1,\cdots, Y_k$. Given this, it appears inappropriate to directly insert $p_{S_Y, S_X}(S_Y, S_X)$ into the integrand, as it is a function dependent on the integrated variables.
3. Even after ignoring the doubts in the proof of Theorem 3, Corollary 1 is questionable. Given Fig.2, how can $S_X$ independent with $Y_{1:k}$ given $S_Y$? In what situation does the "if" statement hold?
4. Could you state the necessary and sufficient conditions for conditional independence consistency in words, given the three equivalent statements in Theorem 3?
5. What does the $n$ mean in Theorem 1? Should it be $k$?
6. In definition 1, it said that "Moreover, the function $\hat{f}$ has some degree of consistency with the underlying causal function $f$." The phrase "some degree of consistency" is ambiguous; could it be explained more precisely?
7. Could you explain how to get the equation $f(X, e_1) + f(0, e_2) = f(0, e_1) + f(X, e_2)$ in the appendix?
8. Can you provide an explanation of how the second equality in $f(X_1,e_1)+f(X_2,e_2) = f_1(X_1)+f_1(X_2)+f_2(e_1)+f_2(e_2) = f_1(X_1 +X_2)+f_2(e_1)+f_2(e_2),$ is derived in the appendix?
9. There is no $\hat{f}$ appearing in the proof of Theorem 1; is it unrelated?

---

> ### Author Response · Authors · 2023-11-20
> **Official Comment by Authors(1/2)**
>
> We are very grateful for the constructive comments and the time devoted to our work.
>
> > [W1,Q2] I doubt that there is a technical flaw in the proof of Theorem 3, which is described in the Questions section.
>
> In the formula derivation, $s_Y$ is not a function of $y_1, \ldots, y_k$. $s_Y$ is a value given at the beginning of the derivation: $p_{S_{Z}|S_{Y},S_{X}}(s_{Z}|s_{Y},s_{X})$, which is completely independent of $y_1, \ldots, y_k$. For the relationship $S_Y = \sum_{i=1}^k Y_i$, yes, it does hold. It is naturally guaranteed by the integrands: $p_{S_{Y},S_{X},Y_{1:k}}(s_{Y},s_{X},y_{1:k})$ (the third equation) and $p_{Y_{1:k}|S_{Y},S_{X}}(y_{1:k}|s_{Y},s_{X})$ (the fourth equation): when $\sum_{i=1}^k y_i \neq s_Y$, both $p_{S_{Y},S_{X},Y_{1:k}}(s_{Y},s_{X},y_{1:k})$ and $p_{Y_{1:k}|S_{Y},S_{X}}(y_{1:k}|s_{Y},s_{X})$ are equal to $0$.
>
>
> > [W2,Q3] The credibility of Corollary 1 is also in question, as it follows Theorem 3. This is mentioned in the Questions section.
>
> > Given Fig.2, how can $S_X$ independent with $Y_{1:k}$ given $S_Y$?
>
> Without the causal faithfulness assumption, it is possible to have conditional independence that does not correspond to d-separation in the DAG. For examples of violations of faithfulness, you can refer to [1].
>
> > In what situation does the "if" statement hold?
>
> Corollary 2 then informs the reader under what conditions the 'if' statement holds.
>
> The logic of Section 4.2 is as follows:
>
> Remark 1: Given the structure of the DAG as shown in Figure 2, chain and fork structures cannot have conditional independence consistency unless there are coincidental circumstances (violations of faithfulness) in the causal functions within the DAG.
>
> Then, where should the violations of faithfulness occur to gain consistency?
>
> Theorem 2 and Corollary 1: The violations of faithfulness should appear in $\{S_X \perp\perp Y_{1:k}\mid S_Y\}$ or $\{S_Z \perp \perp Y_{1:k}\mid S_Y\}$ to gain the consistency.
>
> Regarding the causal functions, how can such conditional independence be established?
>
> Corollary 2: If the causal function between X and Y or the one between Y and Z is linear, then the corresponding conditional independence holds.
>
>
>
>
> > [W3,Q4] While the name of Theorem 3 is "Necessary and Sufficient Condition for Conditional Independence Consistency,"...
>
>
> You may be referring to Theorem 2. Sorry to cause vagueness. The full name of Theorem 2 should be "Necessary and Sufficient Condition for Conditional Independence Consistency of Chain and Fork Structure." We have modified the name to avoid misunderstanding in the revised version. The logic related to this is as follows:
>
>
>
> * We consider three fundamental cases: chain, fork, and collider structures.
> * Under typical assumptions of constraint-based causal discovery, collider structures naturally have conditional independence consistency, but chain and fork structures do not. (Remark 1)
> * Then, under what conditions do chain and fork structures have conditional independence consistency? (See Necessary and Sufficient Condition Theorem 1, Sufficient Conditions Corollaries 1 and 2)
>
> According to the definition of conditional independence consistency, chain and fork structures are supposed to satisfy $S_{X} \perp\perp S_{Z} \mid S_{Y}$(condition (i)), and this is equivalent to conditions (ii) and (iii).
>
>
>
>
> The significance of conditions (ii) and (iii): Although conditions (ii) and (iii) are derived from condition (i) through some simple transformations, they are more specific requirements compared to condition (i). The integrands in (ii) and (iii) are always the multiplication of two parts, one part only related to components X and Y, and the other part only related to components Y and Z. This motivates us to conclude that only by meeting requirements for the substructure can we obtain consistency for the full structure. From this, we claim: partial linearity is sufficient.
>
>
> [1] Weinberger, Naftali. "Faithfulness, coordination and causal coincidences." Erkenntnis 83.2 (2018): 113-133.

---

> > ### Author Response · Authors · 2023-11-20
> > **Official Comment by Authors(2/2)**
> >
> > > [W4] ...did not offer any resolution. Consequently, it is challenging to assess the significance of this paper's contribution.
> >
> >
> >
> > Our contributions are summarized as follows:
> >
> > What is the problem?
> >
> > 'Is the aggregation of causality recoverable?' This is a fundamental question because almost every real-world dataset can be seen to have some degree of aggregation. If recoverability cannot be guaranteed, then the practical application of nonlinear causal discovery would be largely doubtful. However, this issue is largely overlooked. Our paper aims to bring more attention to this issue within the causality community.
> >
> > When and why does this issue occur?
> >
> > Most of our work focuses on this question. We provide both theoretical analysis and simulation experiments to enhance the community's understanding of the nature of this issue.
> >
> > How to solve it?
> >
> > The theoretical results in our paper can motivate potential ways to solve this problem. We make an initial attempt in the original manuscript: if we divide the process of discovering the Markov equivalence class into two steps - skeleton discovery and determining v-structure - Remark 1 in our paper implies that we can always discover the correct v-structure with the skeleton prior given. The experimental results (see Appendix G) show that with a skeleton prior, the results of causal discovery are not damaged by aggregation. Therefore, we believe future work should focus on skeleton discovery or how to obtain the skeleton prior.
> >
> > > [W5] Given Fig.2 and the discussion between 4.2 and Remark 1, the experimental results appear to illustrate that discussion rather than provide evidence for any significant claims in the main paper.
> >
> > Roughly speaking, the main results of Section 4.2 are: 'collider structures always maintain consistency,' and 'partial linearity is sufficient for chain and fork structures to achieve consistency.' Our experimental results regarding conditional independence consistency (Appendix E) do support these claims. The results show that chain and fork structures yield correct results in Linear+Linear, Linear+Nonlinear, and Nonlinear+Linear cases, but produce incorrect results in Nonlinear+Nonlinear cases. For colliders, correct results are obtained in all cases. It is important to note that our experiment randomly selects combinations of different nonlinear functions in each repetition, ensuring our experiments cover a wide range of nonlinear scenarios.
> >
> >
> >
> >
> > > [Q5] What does the n mean in Theorem 1? Should it be k?
> >
> > Yes. Corrected in revised version.
> >
> > > [Q6] In definition 1, it said that "Moreover, the function $\hat{f}$
> >  has some degree of consistency with the underlying causal function f
> > ." The phrase "some degree of consistency" is ambiguous; could it be explained more precisely?
> >
> > Sorry to cause vagueness. We've made it clear in the revised version and rewrite the explanation for Def 1.
> >
> > Modified Def 1:
> >
> > Functional Consistency means that there exists a function $\hat{f}$ such that the temporally aggregated data \(\overline{X}\) is described by $\overline{X} = \hat{f}(\overline{X}, e)$, where $e$ is a noise vector with independent components that are only related to the independent noises encountered in the process, denoted as $e_{2:k+1}$.
> >
> > Explanation:
> >
> > This definition implies that if functional consistency holds, then the aggregated data can at least be represented as a Structural Causal Model (SCM) in vector form, and the source of independent noise aligns with the underlying process. Please note that we allow the generative mechanism $\hat{f}$ to differ from the underlying causal function $f$. However, even with this allowance, achieving functional consistency in the nonlinear case remains challenging, as we will demonstrate in this section.
> >
> >
> > > [Q7,Q8,Q9] Questions for the proof of Theorem 1.
> >
> > We provide a more detailed proof of Theorem 1 in the revised version. Below are our initial responses to your questions.
> >
> > > Could you explain how to get the equation $f(X, e_1) + f(0, e_2) = f(0, e_1) + f(X, e_2)$ in the appendix?
> >
> > According to the definition of functional consistency:$f(X, e_1) + f(0, e_2)=\hat{f}(X,e)$,$f(0, e_1) + f(X, e_2)=\hat{f}(X,e)$, where e is only related to $e_1$, $e_2$.
> >
> > Therefore, $f(X, e_1) + f(0, e_2) = f(0, e_1) + f(X, e_2)=\hat{f}(X,e_1+e_2)$
> >
> >
> > > Can you provide an explanation of how the second equality in $f(X_1,e_1)+f(X_2,e_2) = f_1(X_1)+f_1(X_2)+f_2(e_1)+f_2(e_2) = f_1(X_1 +X_2)+f_2(e_1)+f_2(e_2)$,
> >  is derived in the appendix?
> >
> >
> > $f(X_1,e_1)+f(X_2,e_2) = f_1(X_1)+f_1(X_2)+f_2(e_1)+f_2(e_2)$ is directly derived from the result we obtin before: $f(X,e)=f_1(X)+f_2(e)$
> >
> > And $f(X_1,e_1)+f(X_2,e_2)=f_1(X_1 +X_2)+f_2(e_1)+f_2(e_2)$ is derived from
> > the definition of functional consistency.
> > > There is no $\hat{f}$
> >  appearing in the proof of Theorem 1; is it unrelated?
> >
> > It appears in the detailed proof.

---

> > > ### Comment · Reviewer_766L · 2023-11-22
> > >
> > > I appreciate the authors' clarification; it has effectively corrected my misunderstanding and answered my questions. As a result, the score has been adjusted to a 6.

---

> > > > ### Author Response · Authors · 2023-11-22
> > > >
> > > > Thank you so much for your feedback and for updating the rating. We are very delighted that you kindly found our response helpful.

---

> ### Author Response · Authors · 2023-11-22
> **Look Forward to Your Feedback**
>
> Thank you once again for the time you've dedicated to reviewing our paper. The discussion period will end soon, and we would appreciate the reviewers' feedback. Specifically, if the reviewers have further questions or concerns after reading the rebuttal, we hope to have the opportunity to address them.

---

### Official Review · Reviewer_eLuf · 2023-11-01

**Soundness:** 3 good
**Presentation:** 2 fair
**Contribution:** 3 good
**Rating:** 6
**Confidence:** 4

**Summary:**

In a lot of real world cases, data are actually generated from time-delayed causal relationships, but are reported by aggregating the data by some means. This paper analyses when this aggregation can be a problem for causal discovery algorithms. The authors investigate assumptions on the time-delayed causal relationships which may no longer hold on the aggregated data. As functional and constraint based methods work conditioned on these assumptions, this analysis provides insight into when these methods can fail. These results are corroborated with experiments.

**Strengths:**

- The paper deals with an interesting, important and understudied issue when it comes to causal discovery. This work raises important points for causal discovery with real world data.
- The paper is mostly clear in its exposition, although some sections can be strenghtened.

**Weaknesses:**

- Section 3 is confusing, mainly due to the fact that some terms are a bit vague. While "functional consistency" is defined, it uses words like "compatible" and "degree of consistency" that are vague and confuse the definition. For example, does compatible seems to mean that the aggregated data is a function of the aggregated data and a noise term?  Furthermore, does $\hat{f}$ having some degree of consistency with underlying $f$ mean that they are the same function? As a result of this vagueness, the rest of this section is unclear. A clearer connection to why this consistency is needed would also help the reader a lot.
- I would have liked to see more experiments with various models. For example, only the LiNGAM algorithm is tested, what about ANM? Does the performance degrade for methods that do not make any functional assumptions (e.g. kernel based methods like KCDC [1])?

[1] Mitrovic, Jovana, Dino Sejdinovic, and Yee Whye Teh. "Causal inference via kernel deviance measures." Advances in neural information processing systems 31 (2018).

**Questions:**

- Theorem 1 is confusing to read as its not grammatically correct.
- Why does the infinite K only consider the linear model? Is this due to the previous section? This is not clear.
- Is there a way to validate the assumptions in Theorem 2?

---

> ### Author Response · Authors · 2023-11-20
>
> We are very grateful for the constructive comments and the time devoted to our work.
>
>
> > [W1,Q1] Section 3 is confusing...
>
> Sorry to cause vagueness. We modified Definition 1 and Theorem 1 and rewrite the explanation for Definition 1 in revised version.
>
>
>
> Definition 1:
>
> Functional Consistency means that there exists a function $\hat{f}$ such that the temporally aggregated data \(\overline{X}\) is described by $\overline{X} = \hat{f}(\overline{X}, e)$, where $e$ is a noise vector with independent components that are only related to the independent noises encountered in the process, denoted as $e_{2:k+1}$.
>
> Explanation:
>
> This definition implies that if functional consistency holds, then the aggregated data can at least be represented as a Structural Causal Model (SCM) in vector form, and the source of independent noise aligns with the underlying process. Please note that we allow the generative mechanism $\hat{f}$ to differ from the underlying causal function $f$. However, even with this allowance, achieving functional consistency in the nonlinear case remains challenging, as we will demonstrate in this section.
>
> Theorem 1:
>
> Consider a function $f(X, e)$ that is differentiable with respect to $X$. Define the following:
>
> Statement 1: The function $f$ is of the form $f(x, e) = Ax + f_2(e)$ for some function $f_2$.
>
> Statement 2: For any positive integer $k$, there exists a function $\hat{f}$ such that the functional equation
> $\frac{\sum_{i=1}^k f(X_{i}, e_{i+1})}{g(k)} = \hat{f}(\overline{X}, e)$
> holds for any $X_i$, $e_i$, and any normalization factor $g(k)$, where $e$ is related only to $e_i$ for $i = 2, \dots, k+1$.
>
> Statement 1 is a necessary condition for Statement 2.
>
> > [W2] ...what about ANM? methods that do not make any functional assumptions?
>
> We included experiments on Additive Noise Models (ANM) and methods that do not impose any functional assumptions, such as PC with Kernel-based Conditional Independence (KCI), in our original submission. Please refer to Appendices C, D, and E for details. Figure 8 presents the experimental results for ANM. In our experiments with non-linear models, we consistently employed general score functions for score-based methods and the Kernel Conditional Independence test for constraint-based methods, which are designed to be effective for any non-linear functions.
>
>
>
> > [Q2] Why does the infinite K only consider the linear model?
>
> Why not consider infinite k for nonlinear model?
>
> An infinite k introduces several mathematical challenges in terms of assumption or definition. For instance, the convergence of infinitely aggregated data heavily depends on functions f and g. If f is linear and g(k) = k, then the aggregated data loses randomness due to the law of large numbers. We require $g(k) \sim O(\sqrt{k})$ to maintain a converged variance. Additionally, we must assume data centralization to avoid infinite expectations. If f is nonlinear,(a) an appropriate $g$ that fits $f$ is necessary to maintain a well-defined variance; (b) different variables in the aggregated data may require different $g$ functions; and (c) for some nonlinear $f$, the variance of aggregated data may never converge. In practical scenarios, it is unrealistic to expect that observed data will perfectly align with these requirements of suitable $g$ and centralization. Furthermore, even if we overcome these assumption and definition issues, analyzing the central limit of non-independent and non-identically distributed variables in nonlinear case remains a complex problem in probability theory.
>
> In summary, an infinite k is unnecessary and impractical in real-world scenarios, and addressing convergence issues detracts from our work's fundamental question: "Is aggregation of causality recoverable?".
>
> Why consider an infinite k for the linear model?
>
> The main purpose is to alert the reader to a special issue caused by aggregation: it transforms any distribution into a Gaussian. This issue causes non-Gaussian based causal discovery methods to collapse, even when the functional form itself is not disrupted by aggregation.
>
>
> > [Q3]  ...assumptions in Theorem 2?
>
> Theorem 2 itself does not introduce additional assumptions. If you are referring to the assumptions mentioned in Remark 1, the causal Markov condition and faithfulness assumption are common in causal discovery. To validate the faithfulness assumption, please refer to [1]. In our paper, this means that, given the structure of the DAG as shown in Figure 2, chain and fork structures cannot have conditional independence consistency unless there are coincidental circumstances in the causal functions within the DAG. Remarks 2 and 3 further explain under which specific coincidental conditions the faithfulness assumption might be violated and how chain and fork structures could become conditionally independence consistent.
>
>
> [1] Zhang, Jiji, and Peter Spirtes. "Detection of unfaithfulness and robust causal inference." Minds and Machines 18 (2008): 239-271.

---

> ### Author Response · Authors · 2023-11-22
> **Look Forward to Your Feedback**
>
> Thank you once again for the time you've dedicated to reviewing our paper. The discussion period will end soon, and we would appreciate the reviewers' feedback. Specifically, if the reviewers have further questions or concerns after reading the rebuttal, we hope to have the opportunity to address them.

---

> > ### Comment · Reviewer_eLuf · 2023-11-22
> > **Response**
> >
> > Thanks for your response.
> >
> > The definition of functional consistency still seems incomplete. Is it not required that the condition has to be true for all $X$?
> >
> > I still think some of the presentation could be improved, hence I will keep my score.

---

> ### Author Response · Authors · 2023-11-22
>
> Thanks for your valuable question. We realized that defining the specific mathematical object to which this consistency property applies is crucial. Thus we modified the definitions of functional consistency and conditional independence consistency as below:
>
> Functional Consistency:
>
> Consider an underlying causal process generating temporally aggregated data. This process is said to exhibit functional consistency if there exists a non-trivial function $\hat{f}$, distinct from identity functions, such that for any realization of the states $X_{1:k}$, and the independent noises encountered in the process $e_{2:k+1}$, the temporally aggregated data $\overline{X}$ satisfies the equation $\overline{X} = \hat{f}(\overline{X}, e)$. Here, $e$ denotes a noise vector comprising independent components only depend on $e_{2:k+1}$.
>
> Conditional independence consistency:
>
> Consider an underlying causal process generating temporally aggregated data. This process is said to exhibit conditional independence consistency if the distribution of temporally aggregated data entails a conditional independence set that is consistent with the d-separation set in the summary graph entailed by the original process.
>
>
> It will be incorporated in the future version. Thank you for improving our manuscript. We welcome any further suggestions and feedback you may have!

---

### Official Review · Reviewer_dWzu · 2023-11-04

**Soundness:** 2 fair
**Presentation:** 1 poor
**Contribution:** 2 fair
**Rating:** 5
**Confidence:** 4

**Summary:**

This paper addresses an interesting problem of whether Granger-type causality is consistent with the instantaneous Graphical causal model even after temporal aggregation.

The authors seem to discuss some kind of invariance properties under temporal aggregation, assuming a few typical temporal causal structures, such as the chain and fork.

**Strengths:**

Addresses a fundamental problem of whether different definitions of causality are consistent or not.

**Weaknesses:**

- Many concepts are poorly defined.
- The description is often qualitative and vague. It is hard to follow the line of thought of the authors.

Unfortunately, this paper is not clearly written. Here are instances of vagueness taken from Section 3.1 alone.

- VAR (1) is typically accepted as a linear model. It is not clear if f() covers another model. The terminology seems inconsistent to the notation.
- The number of variables seems to be s but not clearly defined.
- What "k is large" means is not clear.
- $g(k)$ is defined as "any normalization function like g(k)=1". Then, how do you guarantee that $(X_1 -X_{k+1})/g(k)$ vanishes?
- Definition 1 uses undefined terms such as "imply," "compatible," and "some degree of inconsistency." Their definition is not very clear.

**Questions:**

$g(k)$ is defined as "any normalization function like g(k)=1". Then, how do you guarantee that $(X_1 -X_{k+1})/g(k)$ vanishes?

---

> ### Author Response · Authors · 2023-11-20
>
> We are very grateful for the constructive comments and the time devoted to our work.
>
> > [W1] VAR(1) is typically accepted as a linear model. It is not clear if f() covers another model. The terminology seems inconsistent to the notation.
>
> In Section 3.1, our model definition is similar to the definition of a Nonlinear Vector Autoregressive (NVAR) model as described in [1]. However, in our paper, the function f can be either linear or nonlinear. Therefore, we refer to it as a 'general VAR(1)' in Section 3.1. To clarify, we have added some text in the revised version to emphasize that the VAR discussed in our paper encompasses both linear and nonlinear VAR models.
>
>
> > [W2] The number of variables seems to be s but not clearly defined.
>
> Your understanding is correct. We have added additional text in the revised version to clearly define the number of variables as s.
>
> > [W3] What "k is large" means is not clear.
>
> In our paper, we present the sufficient and necessary conditions for two types of consistency in three steps:
>
> *   Define the original model as a time-delay causal model (VAR) and define its temporal aggregated version.
> *   Approximate the aggregation of the time-lag causal model as the aggregation of the instantaneous causal model (at the end of section 3.1 for functional consistency as you mentioned, and section 4.1 for conditional independence consistency).
> *   Present the sufficient and necessary conditions of consistency for the aggregation of the instantaneous model, for any finite positive k, and any choice of g(k).
>
> Among these steps, we only apply the large k condition in the approximation step. Yes, from a mathematical perspective, "k is infinity" would imply a perfect approximation. However, we write "k is large" for the following reasons:
>
>
> 1. According to our experiments, a large but not extremely large(infinity) k is enough to make sure the approximation reasonable. Please see Figure 9 in Appendix F. When k=30, the results of aggregation of time-delay model and the one of instantaneous model become very similar.
>
> 2. An infinite k introduces several mathematical challenges in terms of assumption or definition. For instance, the convergence of infinitely aggregated data heavily depends on functions f and g. If f is linear and g(k) = k, then the aggregated data loses randomness due to the law of large numbers. We require $g(k) \sim O(\sqrt{k})$ to maintain a converged variance. Additionally, we must assume data centralization to avoid infinite expectations. If f is nonlinear,(a) an appropriate
> g that fits f is necessary to maintain a well-defined variance; (b) different variables in the aggregated data may require different g functions; and (c) for some nonlinear f, the variance of aggregated data may never converge. In practical scenarios, it is unrealistic to expect that observed data will perfectly align with these requirements of suitable g and centralization. Furthermore, even if we overcome these assumption and definition issues, analyzing the central limit of non-independent and non-identically distributed variables remains a complex problem in probability theory.
>
> In summary, an infinite k is unnecessary and impractical in real-world scenarios, and addressing convergence issues detracts from our work's fundamental question: "Is aggregation of causality recoverable?". Therefore, we use "large k" in the paper. In the revised version, we have included references to experimental results when mentioning "large k" for additional clarity.
>
>
>
>
>
> > [W4,Q1] g(k) is defined as "any normalization function like g(k)=1". Then, how do you guarantee that $(X_1 -X_{k+1})/g(k)$ vanishes?
>
> We have corrected this in the revised version. Our intent is to convey that in Step 3, when we begin with the aggregation of the instantaneous model and consider any finite k, the choice of g(k) does not matter at all.
>
>
> > [W5] Definition 1 uses undefined terms such as "imply," "compatible," and "some degree of inconsistency." Their definition is not very clear.
>
> We've made it clear in the revised version and rewrite the explaination for Definition 1. Modified Definition 1: Functional Consistency means that there exists a function $\hat{f}$ such that the temporally aggregated data \(\overline{X}\) is described by $\overline{X} = \hat{f}(\overline{X}, e)$, where $e$ is a noise vector with independent components that are only related to the independent noises encountered in the process, denoted as $e_{2:k+1}$.
>
>
> [1] Morioka, Hiroshi, Hermanni Hälvä, and Aapo Hyvarinen. "Independent innovation analysis for nonlinear vector autoregressive process." International Conference on Artificial Intelligence and Statistics. PMLR, 2021.

---

> ### Author Response · Authors · 2023-11-22
> **Looking Forward to Your Feedback**
>
> Thank you once again for the time you dedicated to reviewing our paper.  Please see our response above.  As you probably know, the discussion period will end soon, and we hope for your feedback and the opportunity to respond to it. Could you please kindly let us know whether your comments were properly addressed by our response and whether you have other comments?
>
> Best wishes,
>
> Authors of #7390

---

> ### Comment · Reviewer_dWzu · 2023-11-22
> **Final comment**
>
> There seems to be a significant gap between the description and the intended outcome. For instance, I find the expression $\overline{X}=\hat{f}(\overline{X},e)$ problematic, as the noise component can take any value, implying an identity $\hat{f}(\overline{X},e) = \overline{X}$. The issue may stem from an inadequate definition of underlying stochastic processes, and it appears challenging to rectify. Regrettably, I feel compelled to recommend rejection for this version.

---

> > ### Author Response · Authors · 2023-11-23
> >
> > Sorry for our oversight in Definition 1. Actually, this issue was discussed in previous literature (see Note 2 on page 4 of [1]). In alignment with that paper, we substituted out identities to avoid trivial cases in Definition 1 for the future version. The modified definition 1 is shown below:
> >
> > Consider an underlying causal process generating temporally aggregated data. This process is said to exhibit functional consistency if there exists a non-trivial function $\hat{f}$, distinct from identity functions, such that for any realization of the states $X_{1:k}$, and the independent noises encountered in the process $e_{2:k+1}$, the temporally aggregated data $\overline{X}$ satisfies the equation $\overline{X} = \hat{f}(\overline{X}, e)$. Here, $e$ denotes a noise vector comprising independent components only depend on $e_{2:k+1}$.
> >
> >
> > Please note that this issue does not affect Theorem 1. The statement in Theorem 1 does not involve terms like 'functional consistency' or 'random process'. It's only about a functional equation $\frac{\sum_{i=1}^k f(X_{i}, e_{i+1})}{g(k)} = \hat{f}(\overline{X}, e)$, which is slightly different from the definition of functional consistency. The reason we analyze this statement rather than functional consistency is discussed in the content preceding Theorem 1.
> >
> >
> > [1] Fisher, Franklin M. "A correspondence principle for simultaneous equation models." Econometrica: Journal of the Econometric Society (1970): 73-92.

---

### Author Response · Authors · 2023-11-21
**Revised Manuscript Uploaded**

We sincerely thank the reviewers for their dedicated efforts and the valuable time they invested in carefully reviewing our definitions, theorems, and proofs, as well as their helpful suggestions, which have greatly improved the readability and rigor of our manuscript.

The revised content is shown in blue text. The revised source code has also been updated as supplementary material.

The revised manuscript includes the following changes:

* Reviewer dWzu:

    * clarify the definition of VAR
    * add the definition of $s$
    * add reference to corresponding experiment when mentioning "large k"
    * clarify the choice of g(k)
    * modified Definition 1 and rewrite the explanation

* Reviewer eLuf
    * modified Definition 1 and Theorem 1 and rewrite the explanation for Definition 1
    * rewrite the first paragraph in the experiment section to outline the experiments included

* Reviewer 766L
    * modified the name of Theorem 2
    * add the index for the equations in the proof
    * corrected typo in Theorem 1
    * modified Definition 1 and rewrite the explanation for Definition 1
    * make the proof of Theorem 1 more detailed

* Reviewer 2HMB
    * modified Definition 1 and Theorem 1 and rewrite the explanation for Definition 1
    * clarify the choice of g(k)
    * rewrite the first paragraph in the experiment section to outline the experiments included

* Reviewer yyms
    * rewrite the first paragraph in the experiment section to outline the experiments included

* Others

   * fix a minor bug in the source code of two experiments.
   * redo 2 experiments and replace the results(Table 1 in main text, Table 2 and Figure 9 in Appendix). The corrected results are very similar to the original ones, so they do not affect the content of the paper.
   * make the limitation shorter to meet the page limit.

---

### Meta-Review · Area_Chair_5jUS · 2023-12-06

**Metareview:**

The authors investigate the problem of time aggregation and its role in two different kinds of causal discovery algorithms: functional and constraint-based methods. Although the topic is very practical and deserves attention, multiple reviewers found the notion of functional consistency problematic in the manuscript. The rebuttal was unable to convince all reviewers. After looking at the paper myself, I agree that the assumption (X_1-X_{k-1})/g(k) being completely ignored in the rest of the section is not the best way to proceed. Assuming a compact domain, at the very least the authors should carry over an o(1) term that they then can say vanishes as k goes to infinity. But then the first term 1/g(k)\sum_{i=2}^{k-1}X_i also goes to zero as g(k) goes to infinity. It seems that a more careful analysis is needed to avoid this issue which requires a round of revision.

**Justification For Why Not Higher Score:**

The technical issues need to be addressed on functional consistency before paper can be accepted. I agree with the reviewers that this is an important point related to the soundness of claims.

**Justification For Why Not Lower Score:**

N/A

---

### Decision · Program_Chairs · 2024-01-16

Reject